# STRUCTURED MIXTURE-OF-EXPERTS LLMS COMPRESSION VIA SINGULAR VALUE DECOMPOSITION

## ABSTRACT

Mixture of Experts (MoE) architecture has emerged as a powerful paradigm in the development of Large Language Models (LLMs), offering superior scaling capabilities and reduced computational costs. However, the increased parameter budgets and memory overhead associated with MoE LLMs pose significant challenges to their efficiency and widespread deployment. In this paper, we present MoE-SVD, the first decomposition-based compression framework tailored for MoE LLMs without any extra training. By harnessing the power of Singular Value Decomposition (SVD), MoE-SVD addresses the critical issues of decomposition collapse and matrix redundancy in MoE architectures. Specifically, we first decompose experts into compact low-rank matrices, resulting in accelerated inference and memory optimization. In particular, we propose selective decomposition strategy by measuring sensitivity metrics based on weight singular values and activation statistics to automatically identify decomposable expert layers. Then, we share a single V-matrix across all experts and employ a top-k selection for U-matrices. This low-rank matrix sharing and trimming scheme allows for significant parameter reduction while preserving diversity among experts. Comprehensive experiments conducted on Mixtral-8×7B|22B, Phi-3.5-MoE and DeepSeekMoE across multiple datasets reveal that MoE-SVD consistently outperforms existing compression methods in terms of performance-efficiency tradeoffs. Notably, we achieve a remarkable 60% compression ratio on Mixtral-7x8B and Phi-3.5-MoE, resulting in a 1.5× inference acceleration with minimal performance degradation. Codes are available in the supplementary materials.

## 1 INTRODUCTION

Mixture of Experts (MoE) (Cai et al., 2024b) have demonstrated promising advancements in the realm of large language models (LLMs) (Touvron et al., 2023). These architectures incorporate multiple expert networks and employ a sparse gating mechanism, enabling efficient computation and facilitating the scaling of LLMs within the constraints of limited computational resources(Dai et al., 2024). Numerous studies have shown that MoE models can achieve state-of-the-art results across various benchmarks while utilizing fewer resources compared to traditional dense models (Dai et al., 2024; Fedus et al., 2022; Jiang et al., 2024). Despite these advantages, MoE LLMs still face several challenges: **(1) Immense Parameter Sizes:** MoE models generally have a larger number of parameters than dense models (Xue et al., 2024b), which can make them difficult to train and deploy, especially in resource-constrained environments. **(2) Memory Overhead:** MoE models can suffer from memory inefficiency due to the need to store and access multiple expert weights and biases, potentially hindering their deployment on devices with limited memory (Song et al., 2023).

**Limitations of Traditional MoE Compressions:** To address the challenges of large parameter size and memory overhead, some MoE-specific compression methods have been proposed to prune unimportant experts or weights. For example, Lu et al. (2024) propose task-specific expert pruning and dynamic skipping, while He et al. (2024) evaluate various types of sparse schemes across multiple MoE components. Although these techniques show promise, they suffer from certain limitations: **(1) Performance Degradation,** especially under high compression ratios, often necessitating costly and time-consuming retraining. For instance, pruning 25% of experts in Mixtral-8×7B results in a 23% performance drop (He et al., 2024). **(2) Hardware Dependency:** Some semi-structured sparse methods only gain speedup on NVIDIA Ampere and Hopper architecture GPUs, limiting their

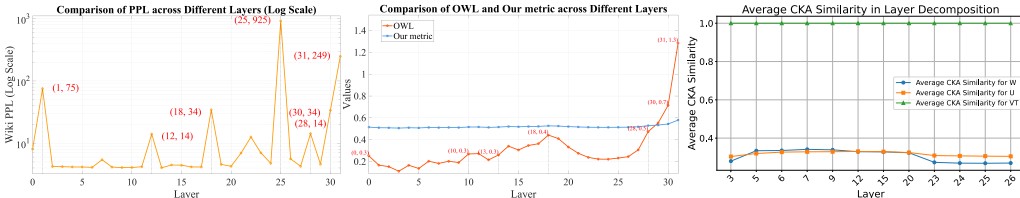

Figure 1: Perplexity of 50% per-layer SVD decomposition (*left*), per-layer values of OWL & our metric (*middle*), mean CKA similarity (Kornblith et al., 2019) of decomposed V & U matrix and original matrix of each expert layer (*right*). TThese results are obtained for Mixtral-8×7B on WikiText-2.

general applicability. **(3) Limited Acceleration:** Some MoE compression methods only reduce the number of experts without significantly reducing the size of the activated experts Liu et al. (2024a), resulting in minimal speedup during inference. He et al. (2024) report that eliminating 12.5% of experts yields less than a 1% speed boost. In contrast, recent Singular Value Decomposition (SVD) techniques (Hsu et al., 2022) are hardware-independent and successfully compact LLM to high compression ratios without additional training. These facts encourage us to explore SVD as an alternative to pruning. However, we directly apply these SVD-based methods to compress MoE models, resulting in serious performance collapse. For example, Mixtral-8×7B with ASVD (Yuan et al., 2023) and SVD-LLM (Wang et al., 2024) reach over 1000 perplexity on WikiText-2. This naturally raises key questions: ***Why SVD-based methods fail on MoE LLMs and how to solve this?***

**Our New Observations:** To answer these questions, we individually decompose each expert layer in Figure 1 and uncover observations: **(1) Decomposition Sensitivity:** Some expert layers are more sensitive to SVD decomposition than others. For example, initial and final layers of decomposition can lead to drastic performance loss. This indicates that layer-wise non-uniform decomposition is important for MoE LLMs. **(2) Model Statistic Disparities:** Activation outliers in OWL (Yin et al., 2023) are an effective pre-layer importance statistic and metric for dense LLM. However, we notice that their values on MoE in Figure 1 (*middle*) do not match the pre-layer decomposition result in (1). This could be attributed to biases derived from multi-expert design and dynamic activation in MoE LLMs. **(3) Expert Redundancy:** Expert merging methods Liu et al. (2024a) show various experts are similar in the weight space and contain significant redundancy. Our Figure 1 (*right*) indicates high similarity of decomposed V-matrices, which can further share weights or trim redundant matrices.

*"Different problems require different solutions."*

— *Albert Einstein*

**Our Novel Framework:** As this well-said quote goes, the sparse-activated MoE dynamic architecture differs from common LLMs and deserves customized decomposition schemes based on the above observations. To this end, we develop MoE-SVD, a novel compression framework specifically designed for MoE LLMs. Our MoE-SVD leverages SVD to decompose expert layers in a structured manner, creating a naturally sparse expert structure that reduces computational costs while maintaining model expressiveness. The core innovation of MoE-SVD lies in: **(1) Selective Decomposition Strategy:** Unlike previous SVD approaches that apply uniform compression across different layers, our method introduces a sensitivity metric derived from matrix singular values and activation statistics, facilitating adaptive decomposition. As illustrated in Figure 1 (*middle*), this metric accurately identifies the sensitive expert layers, allowing for more targeted compression. **(2) Low-rank Matrix Sharing and Trimming:** To further minimize parameter redundancy, we introduce V-matrix sharing, where the most frequently sampled V-matrix is retained and shared across all experts. In addition, we apply U-matrix trimming by selecting the top-k U-matrices based on sampling frequency, while discarding the less frequently used matrices. This strategy significantly minimizes the number of parameters, while ensuring the diversity for effective MoE functioning. With these innovative schemes, our MoE-SVD offers substantial parameter reduction, creates a naturally sparse expert structure for faster inference, and can be deployed on standard computing infrastructure without requiring additional training phases. This flexible framework allows high compression ratios and strikes an optimal balance between computational efficiency and model performance.

**Validation and Results:** Extensive experiments demonstrate Our MoE-SVD method achieves state-of-the-art performance in compressing MoE models while maintaining their performance on

various language modeling and common sense reasoning datasets. The results show that MoE-SVD outperforms other methods such as SVD, ASVD, and SVD-LLM across all compression ratios from 20% to 60%, on both Mixtral-8×7B and Phi-3.5-MoE models. For example, on the Mixtral-8×7B model, MoE-SVD achieves a 20% compression ratio with only a 2% drop in performance, while the other methods experience a significant drop in performance. Similarly, on the Phi-3.5-MoE, MoE-SVD achieves a 40% compression ratio with only a 5% drop in performance. These results demonstrate the effectiveness of MoE-SVD in compressing MoE models while maintaining their performance. In addition, our MoE-SVD can generalize well to other MoE LLMs such as DeepSeekMoE-16B and Mixtral-8×22B, and can be further improved in performance by LoRA fine-tuning and efficiency with quantization. We summarize our contribution as follows:

- To overcome limitations of existing methods, we open new doors for MoE compression from the SVD technical route. We derive series of important findings about decomposition collapse, statistic discrepancies, and redundancy, providing insights into this new area.

- We introduce MoE-SVD, the first SVD-based structured compression method for MoE LLMs. Our MoE-SVD enjoys benefits: high compression ratios, clear inference acceleration, free from specialized hardware and extra training

- We propose a selective decomposition strategy that adaptively applies SVD and develop low-rank matrix sharing and trimming techniques. By sharing V-matrices across experts and trimming redundant U-matrices, we achieve significant parameter reduction while maintaining expert diversity and model performance.

- Extensive experiments demonstrate the effectiveness of MoE-SVD on Mixtral-8×7B|22B, Phi-3.5-MoE and DeepSeekMoE. Our MoE-SVD consistently outperforms other SVD-based methods across 20% ∼ 60% compression ratios and achieves $1.2\times \sim 1.5\times$ inference speedups.

## 2 RELATED WORK

**Mixture of Experts.** MoE is initially introduced for conditional computation (Jacobs et al., 1991; Jordan & Jacobs, 1994; Eigen et al., 2013) and has evolved into a sparse activation framework (Shazeer et al., 2017) that enables efficient training and inference in language (Fedus et al., 2022) tasks. The architecture's ability to achieve superior scaling laws at reduced costs (Clark et al., 2022) has led to its widespread adoption in state-of-the-art Large Language Models (LLMs) (Jiang et al., 2024; Dai et al., 2024). Recent advancements in MoE focus on refining expert structures (Rajbhandari et al., 2022; Dai et al., 2024), enhancing router designs (Zhou et al., 2022; Zoph et al., 2022), and developing innovative training strategies (Chen et al., 2022; Liu et al., 2023). However, MoE LLMs still face challenges, including increased parameter budgets due to expert replication (He et al., 2023), communication costs that enhance latency (Song et al., 2023; Xue et al., 2024b), and significant memory overhead issues (Li et al., 2024b), posing challenges to their efficiency.

**MoE Compression.** To address the above issues, researchers are developing MoE-specific compression methods (*e.g.*, expert pruning (Lu et al., 2024; He et al., 2024)). For instance, Liu et al. (2024a) proposes search-based expert pruning and merging, while Zhang et al. (2024) investigates task-agnostic pruning methods that diversify expert knowledge. In contrast, our MoE-SVD develops a new decomposition route: (1) Unlike other methods that require training and drop inactive experts without acceleration, MoE-SVD primarily exploits SVD to reduce the size of activated experts for acceleration without extensive retraining. (2) MoE-SVD performs low-rank matrix sharing and trimming, avoiding the direct dropping of entire experts like other methods, which prevents drastic performance loss. (3) Other approaches include post-training quantization (Li et al., 2024a) and system optimization (*e.g.*, expert offloading (Xue et al., 2024a), parallelism (Cai et al., 2024a), and switching (Liu et al., 2024b)). Our MoE-SVD focuses solely on expert compression and remains orthogonal to these methods. (4) Our MoE-SVD enhances the SVD for large-scale MoEs without expert merging and fine-tuning, in contrast to MC-SMoE (Li et al., 2024b), which only addresses small-scale MoE by first merging experts and then applying vanilla decomposition before fine-tuning.

**Singular Value Decomposition.** Recently, several SVD-based methods have been proposed for compressing LLMs (Golub et al., 1987). For example, FWSVD (Hsu et al., 2022) introduces a weighted low-rank factorization, while ASVD (Yuan et al., 2023) proposes an activation-aware SVD method that considers the activation patterns of the model's layers to improve compression

Figure 2: Pipeline of MoE-SVD. We first selectively decompose expert layers with SVD, dividing them into U and V matrices. Then, we present V-matrix sharing and U-matrix trimming steps. For V-matrix sharing, we retain only a single V-matrix $V_s$ and share it across experts. For U-matrix trimming, we perform frequency-based top-k selection of U-matrices and trim out unselected ones.

efficiency. Meanwhile, SVD-LLM (Wang et al., 2024) adopts truncation-aware data whitening and layer-wise parameter update strategies to achieve better compression ratios. In contrast to these general SVD-based methods, our MoE-SVD is specifically designed for MoE LLMs, addressing their unique challenges (*e.g.*, decomposition sensitivity and expert redundancy). Additionally, while these methods typically uniformly decompose every layer in LLMs, our method employs adaptive decomposition across various expert layers in MoE LLMs.

## 3 METHODOLOGY

Our MoE-SVD introduces SVD expert decomposition, selective decomposition strategy, low-rank matrix sharing and trimming to reduce model parameters while maintaining performance. The main process of MoE-SVD is illustrated in Figure 2. More algorithm details are in Appendix C.

### 3.1 SVD EXPERT DECOMPOSITION IN MOE LLMS

**Recap of MoE Formulation.** MoE architectures in LLM enhance model capacity and efficiency by using expert-based Feed-Forward Network (FFN) layers for different input tokens . The output $y$ of the MoE-FFN layer for input $x$ is computed as:

$$y = \sum_{i=1}^{N} G(x)_i \cdot E_i(x),\tag{1}$$

where $N$ is the number of experts, $G(x)$ is the gating function, The gating function $G(x)$ typically employs a top-$k$ selection mechanism, where only the top-k experts are activated $G'(x) = \text{TopK}(G(x), k)$, resulting in a sparse output. and $E_i(x)$ is the output of the $i$-th expert. Each expert $E_i$ is a standard FFN with two or three fully-connected layers. These FFN experts take most of the parameters and memory overhead in MoE models. Therefore, our method and other MoE compression methods are concerned with this part of the compression.

**SVD-based Expert Decomposition.** In our SVD-based framework, we apply SVD to decompose the weights of expert layers. Consider an MoE model with $N$ experts, each fully-connected layer represented by a weight matrix $W_i \in \mathbb{R}^{m \times n}$, where $i \in \{1, ..., N\}$. We begin by applying SVD to each expert matrix:

$$W_i = U_i \Sigma_i V_i^T,\tag{2}$$

where $U_i \in \mathbb{R}^{m \times m}$ and $V_i \in \mathbb{R}^{n \times n}$ are orthogonal matrices containing the left and right singular vectors, respectively, and $\Sigma_i \in \mathbb{R}^{m \times n}$ is a diagonal matrix containing the singular values in descending order, respectively. To create a sparse MoE structure, we first factorize each expert $\mathbf{W}$ using the SVD decomposition and then trunca the top-k singular values and their corresponding singular vectors and finally reconstruct an approximated weight matrix:

$$\{E_i\}_{i=1}^{k} = \{u_i \cdot \sigma_i \cdot v_i^T\}_{i=1}^{k},\tag{3}$$

where $u_i$ and $v_i$ are the $i$-th columns of $U$ and $V$ respectively, and $\sigma_i$ is the $i$-th singular value. Following ASVD, we also address activation outliers by scaling the weight matrix based on the activation distribution, enhancing decomposition accuracy (see more details in Appendix D.2 D). Our SVD-based expert decomposition creates a naturally sparse expert structure, potentially reducing computational costs. The number of experts can be easily adjusted, allowing for fine-grained control over the MoE's capacity and computational requirements, avenues for compress MoE LLMs.

## 3.2 SELECTIVE DECOMPOSITION STRATEGY

To determine the sensitivity of expert layers in the MoE to decomposition, we employ a selective decomposition strategy. This approach is based on a carefully crafted sensitivity metric that considers both the singular value decomposition of expert weight matrices and the activation patterns of these experts during inference. For a layer with $N$ experts, we normalize these sensitivities using expert sampling frequency to obtain the layer-wise sensitivity metric $S_L$:

$$S_L = \sum_{i=1}^{N} f_i \cdot r_i \cdot a_i, \tag{4}$$

where $f_i$ represents the sampling frequency of the $i$-th expert during router selection, $r_i$ denotes the principal rank (number of large value components) of singular vectors $\Sigma_i = \mathrm{diag}(\sigma_{i,1}, \sigma_{i,2}, ..., \sigma_{i,d})$ obtained from the SVD of the $i$-th expert's weight matrix, and $a_i$ measures the proportion of activation outliers of the $i$-th expert exceeding the mean absolute activation value (see more details in Appendix D.1 D). To apply selective decomposition, we set a threshold $\tau$ based on the desired compression ratio. Expert layer with sensitivity $S_i \geq \tau$ are preserved without decomposition, while those below the threshold undergo SVD decomposition. This process is repeated for each layer in the network. This selective decomposition strategy allows for a nuanced approach to MoE compression, preserving the most important experts while reducing the computational footprint of less critical components.

Table 1: Decomposed layers of our selective decomposition strategy by varying compression ratios.

| | Mixtral-8×7B | | Phi3.5 MoE |
| --- | --- | --- | --- |
| Ratios | Selected Decomposition Expert Layers | Ratios | Selected Decomposition Expert Layers |
| 20 | [3,5,6,7,9,12,23,24,25], | 20 | [11,12,15,20,21,23,24,25], |
| 30 | [3,5,6,7,9,12,13,22,23,24,25,26], | 30 | [11,12,15,20,21,23,24,25,27,28], |
| 40 | [3,5,6,7,9,10,12,13,21,22,23,24,25,26], | 40 | [10,11,12,15,16,18,20,21,23,24,25,26,27,28], |
| 50 | [3,5,6,7,9,10,12,13,14,15,16,20,21,22,23,24,25,26], | 50 | [5,6,10,11,12,13,15,16,18,20,21,23,24,25,26,27,28], |
| 60 | [2,3,5,6,7,9,10,12,13,14,15,16,17,20,21,22,23,24,25,26,27] | 60 | [5,6,10,11,12,13,15,16,18,19,20,21,23,24,25,26,27,28,29] |

**Decomposable Expert Layer Analysis.** To better understand our selective decomposition, we show layer decomposition results for Mixtral-8×7B and Phi-3.5-MoE in Table 1. As the compression increases from 20% to 60%, both models exhibit a gradual increase in decomposed layers, albeit with distinct characteristics. Mixtral-8×7B displays a more aggressive decomposition pattern, with approximately half of its layers decomposed at 60% compression, whereas Phi-3.5-MoE demonstrates greater resilience, maintaining more undecomposed layers at higher compression ratios. Notably, both models consistently undecompose their initial and final layers across all compression levels, suggesting the critical nature of these layers for maintaining model performance. In contrast, certain middle layers in both architectures show remarkable tendencies to decomposition. Mixtral-8×7B exhibits a block-like decomposition pattern, while Phi-3.5-MoE showcases a more uniform distribution of decomposed layers. These observations reveal intriguing patterns and present insights for our selective decomposition of expert layers under layer-wise MoE compression.

## 3.3 LOW-RANK MATRIX SHARING AND TRIMMING

**Motivation:** For MoE models, different expert matrices contain some similarities and can be merged Liu et al. (2024a). As shown in Figure 1 (*right*), decomposed V-matrices share certain similarities. Consider two expert matrices $W_1$ and $W_2$ with SVD decompositions $W_1 = U_1 \Sigma_1 V_1^T$ and $W_2 = U_2 \Sigma_2 V_2^T$. The similarity in their output transformations can be quantified by the Frobenius inner product of their V-matrices $\langle V_1, V_2 \rangle_F = \mathrm{tr}(V_1^T V_2)$. For experts trained on similar tasks, this inner product is often close to high values, indicating high similarity in output transformations. Consequently, we can perform a more fine-grained matrix selection and sharing, achieving a superior trade-off between performance and the number of parameters.

**V-matrix Sharing:** We compress MoE by retaining only the V-matrix with the highest router sampling frequency and sharing this matrix across all experts. This method significantly reduces the model's memory footprint while preserving crucial directional information in the feature space. The router sampling frequency $f(V_i)$ for each expert $i$ is computed based on the routing decisions made by the gating network $G(x)$. The shared V-matrix, denoted as $V_s$, is selected as follows:

$$V_s = \arg\max_{V_i} f(V_i), \quad f(V_i) = \frac{\sum_{x \in \mathcal{X}} \mathbb{I}[i \in \mathrm{TopK}(G(x), k)]}{|\mathcal{X}|}, \tag{5}$$

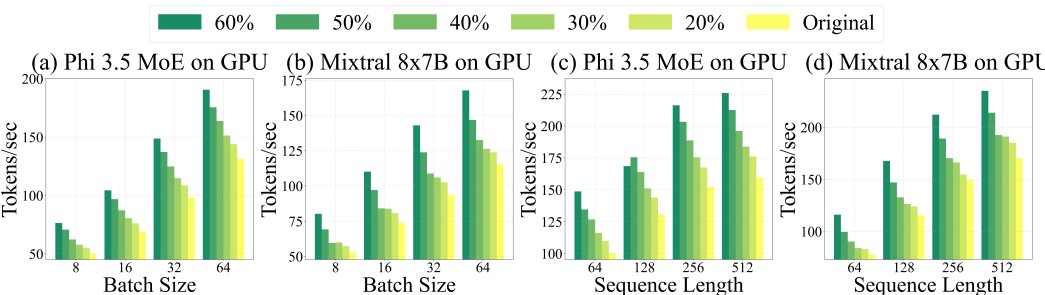

Figure 3: Throughput (Tokens/sec) of Mixtral-8×7B and Phi-3.5-MoE compressed by MoE-SVD at 20%∼60% ratios on a single H800 GPU is compared in Figures (a) & (b) for various batch sizes at sequence length = 32, and in Figures (c) & (d) for varying sequence lengths at batch size = 64.

where $\mathcal{X}$ represents the set of all input tokens, $\mathbb{I}[\cdot]$ is the indicator function, and $k$ denotes the number of experts selected by the top-k gating mechanism. After selecting $V_s$, we update all expert matrices to use this shared V-matrix:

$$E_i \approx U_i \Sigma_i V_s^T, \quad i = 1, \ldots, N, \tag{6}$$

The shared $V_s$ matrix encapsulates common output space transformations across all experts. For the expert matrix $W_i$ using the shared V-matrix $V_s$, its expected reconstruction error is $\mathbb{E}[\|W_i - \tilde{W}_i\|_F^2] = \mathbb{E}[\|W_i - U_i \Sigma_i V_s^T\|_F^2]$. Minimizing this error is equivalent to maximizing the correlation between $W_i$ and $U_i \Sigma_i V_s^T$. Given that $V_s$ is chosen based on the highest router sampling frequency, it represents the most commonly used output transformation. Therefore, sharing $V_s$ minimizes the expected reconstruction error across all experts.

**U-matrix Trimming:** For the remaining U-matrices, we employ a top-k selection strategy based on router sampling frequency. The diversity among experts is primarily maintained through the unique $U_i \Sigma_i$ components. Typically, we set $k = 2$ to balance parameter efficiency and expert diversity. The selected U-matrices for each expert are determined as follows:

$$\{U_{i,1}, U_{i,2}\} = \text{TopK}(\{U_j | f(V_j) > f(V_i)\}, k = 2), \quad i = 1, \ldots, N, \tag{7}$$

where TopK selects the $k$ U-matrices with the highest router sampling frequencies among those experts more frequently sampled than expert $i$. The final expert function for expert $i$ becomes:

$$E_i(x) = (U_{i,1}\Sigma_{i,1} + U_{i,2}\Sigma_{i,2})V_s^T x. \tag{8}$$

where $\Sigma_{i,1}$ and $\Sigma_{i,2}$ are the corresponding singular value matrices for the selected U-matrices.
**Parameter Reduction:** Our method achieves significant parameter reduction compared to the original MoE model with $N \times m \times n$ parameters. After applying our low-rank decomposition, each expert matrix $W_i$ is decomposed into $U_i\Sigma_i \in \mathbb{R}^{m \times r}$, $\Sigma_i \in \mathbb{R}^{r \times r}$, and $V_i \in \mathbb{R}^{r \times n}$, resulting in $N \times (m \times r + r \times n)$ parameters per expert. With V-matrix sharing, we retain only one $V_s \in \mathbb{R}^{r \times n}$ matrix shared across all experts, reducing the parameter count by a factor of $N$ for the V-matrices. Furthermore, with U-matrix selection, we typically select $k = 2$ U-matrices, reducing the parameter count by a factor of $\frac{N}{2}$ for the U-matrices. Thus, the total number of parameters in our compressed MoE model is $m \times r \times 2 + r \times n = 2mr + rn$. Comparing this to the original $N \times m \times n$ parameters, we achieve a substantial reduction in the number of parameters, especially when $r \ll \min(m, n)$. The parameter reduction ratio is approximately $\frac{2mr+rn}{Nmn}$, which can be significant for LLMs with high input and output dimensions.

**Experts Diversity:** Despite the parameter reduction, our method still maintains diversity among experts by leveraging the unique components of $U_i\Sigma_i$: the $U$ matrices capturing expert-specific input space transformations and the $\Sigma_i$ matrices determining the importance of these transformations. Through selecting distinct combinations of $U$ matrices for each expert, individual transformations of the input space are preserved, ensuring uniqueness across experts. This approach allows for a balance between parameter efficiency and the preservation of expert diversity, crucial for the effective functioning of the MoE architecture.

Table 2: Zero-shot performance of MoE-SVD and other SVD-based methods for Mixtral-8×7B and Phi-3.5-MoE on three language modeling datasets (measured by perplexity (↓)) and seven common sense reasoning datasets (measured by both individual and average accuracy (↑)).

| Ratio | Method | WikiText-2↓ | PTB↓ | C4↓ | Openb. | ARC_e | WinoG. | HellaS. | ARC_c | PIQA | MathQA | Average↑ |
|---|---|---|---|---|---|---|---|---|---|---|---|---|
| | | | | | **Mixtral-8×7B** | | | | | | | |
| 0% | Original | 3.98 | 12.99 | 6.78 | 0.36 | 0.84 | 0.76 | 0.65 | 0.57 | 0.82 | 0.43 | 0.63 |
| 20% | SVD | 18.80 | 80.04 | 23.92 | 0.24 | 0.57 | 0.58 | 0.43 | 0.33 | 0.68 | 0.23 | 0.44 |
| | ASVD | 9.44 | 47.29 | 20.30 | 0.25 | 0.71 | 0.66 | 0.48 | 0.40 | 0.73 | 0.35 | 0.51 |
| | SVD-LLM | 13.45 | 42.72 | 17.36 | 0.22 | 0.62 | 0.58 | 0.42 | 0.29 | 0.71 | 0.26 | 0.44 |
| | **MoE-SVD** | **5.94** | **19.42** | **8.98** | **0.28** | **0.75** | **0.69** | **0.55** | **0.45** | **0.78** | **0.36** | **0.55** |
| 30% | SVD | 41.62 | 305.29 | 49.59 | 0.20 | 0.44 | 0.55 | 0.36 | 0.25 | 0.63 | 0.23 | 0.38 |
| | ASVD | 17.29 | 79.60 | 30.63 | 0.21 | 0.63 | 0.60 | 0.30 | 0.34 | 0.68 | 0.32 | 0.44 |
| | SVD-LLM | 32.84 | 95.82 | 67.59 | 0.23 | 0.62 | 0.59 | 0.32 | 0.30 | 0.71 | 0.26 | 0.43 |
| | **MoE-SVD** | **6.69** | **20.61** | **9.84** | **0.27** | **0.72** | **0.69** | **0.52** | **0.43** | **0.76** | **0.32** | **0.53** |
| 40% | SVD | 1771.53 | 9069.01 | 3429.04 | 0.15 | 0.30 | 0.52 | 0.27 | 0.22 | 0.53 | 0.19 | 0.31 |
| | ASVD | 30.57 | 196.02 | 87.74 | 0.18 | 0.41 | 0.58 | 0.34 | 0.22 | 0.59 | 0.22 | 0.36 |
| | SVD-LLM | 254.76 | 252.25 | 79.40 | 0.16 | 0.43 | 0.52 | 0.33 | 0.22 | 0.63 | 0.23 | 0.36 |
| | **MoE-SVD** | **8.66** | **27.73** | **12.41** | **0.22** | **0.66** | **0.67** | **0.47** | **0.34** | **0.71** | **0.28** | **0.48** |
| 50% | SVD | 5381.79 | 6320.66 | 6653.16 | 0.13 | 0.27 | 0.50 | 0.25 | 0.21 | 0.51 | 0.20 | 0.30 |
| | ASVD | 86.61 | 402.60 | 164.57 | 0.15 | 0.42 | 0.53 | 0.30 | 0.22 | 0.61 | 0.22 | 0.35 |
| | SVD-LLM | 1325.51 | 1856.62 | 439.20 | 0.14 | 0.33 | 0.48 | 0.28 | 0.21 | 0.56 | 0.23 | 0.32 |
| | **MoE-SVD** | **12.37** | **42.93** | **16.18** | **0.20** | **0.57** | **0.57** | **0.40** | **0.29** | **0.68** | **0.25** | **0.42** |
| 60% | SVD | 3795.00 | 13767.78 | 10037.77 | 0.13 | 0.27 | 0.50 | 0.26 | 0.22 | 0.53 | 0.20 | 0.30 |
| | ASVD | 12524.91 | 14702.02 | 11691.72 | 0.13 | 0.26 | 0.51 | 0.26 | 0.21 | 0.53 | 0.21 | 0.30 |
| | SVD-LLM | 10181.25 | 9284.95 | 10987.80 | 0.14 | 0.26 | 0.51 | 0.26 | 0.22 | 0.54 | 0.21 | 0.30 |
| | **MoE-SVD** | **33.24** | **133.98** | **41.72** | **0.15** | **0.43** | **0.51** | **0.32** | **0.22** | **0.62** | **0.24** | **0.36** |
| | | | | | **Phi-3.5-MoE** | | | | | | | |
| 0% | Original | 3.48 | 8.43 | 8.22 | 0.40 | 0.77 | 0.76 | 0.68 | 0.56 | 0.79 | 0.38 | 0.62 |
| 20% | SVD | 7.18 | 13.38 | 10.42 | 0.37 | 0.70 | **0.74** | 0.59 | 0.52 | 0.75 | 0.35 | 0.57 |
| | ASVD | 7.22 | **10.66** | 9.58 | 0.35 | 0.73 | 0.72 | 0.57 | 0.49 | **0.75** | 0.34 | 0.56 |
| | SVD-LLM | 8.34 | 14.77 | 12.89 | 0.31 | 0.67 | 0.66 | 0.53 | 0.45 | 0.72 | 0.22 | 0.51 |
| | **MoE-SVD** | **4.77** | 12.12 | **9.56** | **0.39** | **0.77** | 0.69 | **0.59** | **0.53** | 0.74 | **0.35** | **0.58** |
| 30% | SVD | 9.95 | 16.18 | 13.89 | **0.34** | 0.64 | 0.65 | 0.45 | 0.46 | 0.69 | 0.34 | 0.51 |
| | ASVD | 9.06 | 15.34 | 14.11 | 0.32 | 0.72 | 0.69 | 0.49 | 0.46 | 0.71 | 0.30 | 0.52 |
| | SVD-LLM | 14.47 | 24.04 | 17.77 | 0.29 | 0.60 | 0.66 | 0.48 | 0.41 | 0.69 | 0.22 | 0.48 |
| | **MoE-SVD** | **5.41** | **13.41** | **10.54** | 0.31 | **0.74** | **0.70** | **0.55** | **0.48** | **0.73** | **0.34** | **0.55** |
| 40% | SVD | 38.83 | 68.52 | 43.81 | 0.23 | 0.56 | 0.60 | 0.39 | 0.31 | 0.66 | 0.24 | 0.43 |
| | ASVD | 14.51 | 22.14 | 21.82 | **0.30** | 0.69 | 0.63 | 0.41 | 0.40 | 0.68 | **0.25** | 0.48 |
| | SVD-LLM | 6494.87 | 6451.79 | 9348.47 | 0.16 | 0.29 | 0.49 | 0.27 | 0.23 | 0.53 | 0.20 | 0.31 |
| | **MoE-SVD** | **6.86** | **16.93** | **13.71** | 0.29 | **0.72** | **0.66** | **0.49** | **0.45** | **0.71** | 0.22 | **0.51** |
| 50% | SVD | 343.14 | 654.54 | 623.97 | 0.18 | 0.46 | 0.55 | 0.32 | 0.25 | 0.60 | 0.21 | 0.37 |
| | ASVD | 20.58 | 33.53 | 30.26 | 0.23 | 0.62 | 0.62 | 0.36 | 0.32 | 0.66 | **0.24** | 0.44 |
| | SVD-LLM | 6494.87 | 6451.79 | 9348.47 | 0.16 | 0.29 | 0.49 | 0.27 | 0.23 | 0.53 | 0.20 | 0.31 |
| | **MoE-SVD** | **8.10** | **21.44** | **18.47** | **0.27** | **0.68** | **0.64** | **0.44** | **0.41** | **0.69** | 0.23 | **0.48** |
| 60% | SVD | 15489.73 | 9886.27 | 10088.20 | 0.12 | 0.26 | 0.51 | 0.26 | 0.22 | 0.52 | 0.19 | 0.30 |
| | ASVD | 107.71 | 208.69 | 161.40 | 0.18 | 0.40 | 0.53 | 0.30 | 0.24 | 0.59 | **0.23** | 0.35 |
| | SVD-LLM | 7168.09 | 7101.49 | 7119.43 | 0.15 | 0.28 | 0.51 | 0.26 | 0.22 | 0.54 | 0.21 | 0.31 |
| | **MoE-SVD** | **12.71** | **36.60** | **30.38** | **0.23** | **0.57** | **0.60** | **0.39** | **0.33** | **0.67** | 0.21 | **0.43** |

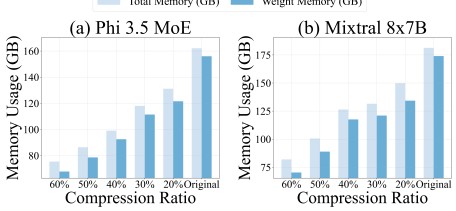

Figure 4: Memory usage (GB) of MoE-SVD for Mixtral-8×22B (a) and Phi-3.5-MoE (b) at varying compression ratios.

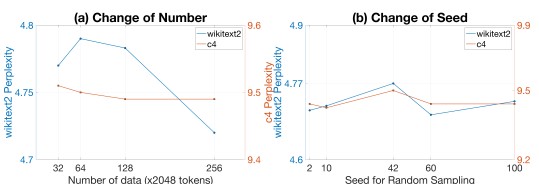

Figure 5: Perplexity of 20% compressed Mixtral-8×22B via calibration data with varying number (a) and seeds (b) from WikiText-2 and C4.

## 4 EXPERIMENTS

In this section, we first compare MoE-SVD against vanilla SVD and state-of-the-art SVD-based methods (*e.g.*, ASVD and SVD-LLM) on Mixtral-8×7B and Phi-3.5-MoE at different compression ratios.

Table 3: Performance of different decomposition settings on Mixtral-8×7B.

| Method | WikiText-2↓ | PTB↓ | C4↓ | Openb. | ARC_e | WinoG. | HellaS. | ARC_c | PIQA | MathQA | Average↑ |
|---|---|---|---|---|---|---|---|---|---|---|---|
| Original | 3.98 | 12.99 | 6.78 | 0.36 | 0.84 | 0.76 | 0.65 | 0.57 | 0.82 | 0.43 | 0.63 |
| Uniform SVD | 18.80 | 80.04 | 23.92 | 0.24 | 0.57 | 0.58 | 0.43 | 0.33 | 0.68 | 0.23 | 0.44 |
| Non-uniform SVD (OWL) | 16.57 | 62.13 | 30.82 | 0.26 | 0.61 | 0.64 | 0.45 | 0.32 | 0.68 | 0.29 | 0.46 |
| Non-uniform SVD (Our selective decompose) | 8.67 | 26.72 | 12.06 | 0.24 | 0.67 | 0.66 | 0.48 | 0.35 | 0.72 | 0.28 | 0.49 |
| Non-uniform SVD (Our selective decompose+trimming) | 5.94 | 19.42 | 8.98 | 0.28 | 0.75 | 0.69 | 0.55 | 0.45 | 0.78 | 0.36 | 0.55 |

Table 4: Perplexity (↓) performance of our MoE-SVD with various numbers of trimmed matrices for Mixtral-8×7B on WikiText-2.

| U-matrix trimming | 1 | 2 | 3 | 4 | 5 | 6 | 7 |
|---|---|---|---|---|---|---|---|
| Perplexity | 10.21 | 9.88 | 9.15 | 8.59 | 8.13 | 6.34 | 7.31 |

Then, we conduct ablation studies and extend MoE-SVD with LoRA fine-tuning and quantization. All experiments are conducted on NVIDIA H800 GPUs.

## 4.1 EXPERIMENTAL SETUPS

**Models and Datasets.** To showcase the versatility of our MoE-SVD method, we assess its effectiveness on Mixtral models (8×7B and 8×22B), Phi-3.5-MoE, and DeepSeek-MoE. Mixtral variations employ 8 experts, achieving remarkable language modeling capabilities. Phi-3.5-MoE excels with 16×3.8 B parameters, while DeepSeek-MoE, with 16 B parameters utilizing fine-grained experts, also exhibits superior performance. We evaluate our method across 10 datasets, encompassing 3 language modeling datasets (WikiText-2 (Merity et al., 2017), PTB (Marcus et al., 1993), and C4 (Raffel et al., 2020)), along with 7 common sense reasoning datasets (OpenbookQA (Mihaylov et al., 2018), WinoGrande (Sakaguchi et al., 2020), HellaSwag (Zellers et al., 2019), PIQA (Bisk et al., 2020), MathQA (Amini et al., 2019), ARC-e, and ARC-c (Clark et al., 2018)) in a zero-shot setting using the LM-Evaluation-Harness framework (Gao et al., 2023).

**Implementation Details.** For fair comparisons, we followed the same settings as ASVD and SVD-LLM and used 256 random samples from WikiText-2 as calibration data. We focus on compressing the model without retraining the full model parameters. See Appendix D for more details.

## 4.2 PERFORMANCE AND ACCELERATION RESULTS

**Performance Comparisons.** The experimental results in Table 2 demonstrate the effectiveness of our MoE-SVD method across various compression ratios for both Mixtral-8×7B and Phi-3.5-MoE models. For Mixtral-8×7B, MoE-SVD consistently outperforms baseline methods (SVD, ASVD, and SVD-LLM) across all compression ratios. At 20% compression, MoE-SVD achieves a WikiText-2 perplexity of 5.94, compared to 18.80 for SVD, 9.44 for ASVD, and 13.45 for SVD-LLM. This trend continues for higher compression ratios, with MoE-SVD maintaining significantly lower perplexities on all language modeling datasets. In terms of common sense reasoning tasks, MoE-SVD maintains higher average accuracies across compression ratios. At 20% compression, our method achieves 0.55 average accuracy, compared to 0.44 for SVD and SVD-LLM, and 0.51 for ASVD. This performance gap widens at higher compression ratios, with MoE-SVD retaining 0.42 average accuracy even at 50% compression, while other methods drop below 0.35. For Phi-3.5-MoE, the performance trends are similar, albeit with smaller margins. MoE-SVD still outperforms baselines in most scenarios, particularly at higher compression ratios. At 20% compression, MoE-SVD achieves a WikiText-2 perplexity of 4.77, slightly better than ASVD (5.22) and significantly better than SVD (7.18) and SVD-LLM (8.34). Notably, MoE-SVD's performance degrades more gracefully as compression increases. At 60% compression for Mixtral-8×7B, MoE-SVD maintains a WikiText-2 perplexity of 33.24, while other methods exceed 3000. Similarly, for common sense tasks, MoE-SVD retains 0.36 average accuracy and surpasses other methods. These results highlight MoE-SVD's robustness and effectiveness in preserving model performance across various tasks and compression ratios, demonstrating its potential for efficient model compression in MoE architectures.

**Inference Speed Acceleration.** Figure 3 demonstrates significant hardware inference acceleration across various batch sizes and sequence lengths for both Phi-3.5-MoE and Mixtral-8×7B models. As the compression ratio increases, a clear trend of improved acceleration emerges, with the most

Table 5: Performance of Mixtral-8×7B compressed by MoE-SVD under 20% compression ratios using calibration data randomly sampled from WikiText-2 (by default in our paper) and C4.

| Calibration | WikiText-2↓ | PTB↓ | C4↓ | Openb. | ARC_e | WinoG. | HellaS. | ARC_c | PIQA | MathQA | Average↑ |
|---|---|---|---|---|---|---|---|---|---|---|---|
| WikiText-2 | 4.77 | 12.12 | 9.56 | 0.39 | 0.77 | 0.69 | 0.59 | 0.53 | 0.74 | 0.35 | 0.58 |
| C4 | 4.82 | 12.15 | 9.60 | 0.34 | 0.72 | 0.70 | 0.59 | 0.48 | 0.74 | 0.26 | 0.55 |

Table 6: Zero-shot performance (average accuracy (↑)) of DeepSeekMoE-16B and Mixtral-8×22B with 20% compression ratio on reasoning datasets.

| Models | Openb. | ARC_e | WinoG. | HellaS. | ARC_c | PIQA | MathQA | Average↑ |
|---|---|---|---|---|---|---|---|---|
| DeepSeekMoE-16B Original | 0.33 | 0.76 | 0.71 | 0.58 | 0.44 | 0.79 | 0.31 | 0.56 |
| DeepSeekMoE-16B MoE-SVD | 0.20 | 0.52 | 0.57 | 0.50 | 0.27 | 0.66 | 0.24 | 0.42 |
| Mistral-8x22B Original | 0.37 | 0.86 | 0.81 | 0.67 | 0.86 | 0.83 | 0.51 | 0.70 |
| Mistral-8x22B MoE-SVD | 0.30 | 0.75 | 0.75 | 0.58 | 0.50 | 0.77 | 0.38 | 0.57 |

substantial gains observed at higher compression levels. For Phi-3.5-MoE, the acceleration ratio peaks at $1.52 \times$ faster than the original MoE with a 60% compression ratio and a batch size of 8. Mixtral-8×7B exhibits similar performance improvements, reaching a maximum acceleration of $1.53 \times$ at 60% compression with a batch size of 32. Notably, the acceleration benefits are consistently observed across different sequence lengths, with both models showing enhanced performance even for longer sequences. These results underscore the practical value of MoE-SVD in achieving tangible speedups for LLM inference, potentially enabling more efficient deployment of these models in resource-constrained environments while maintaining a significant portion of their original capabilities.

**Memory Reduction Analysis.** Figure 8 unveils the remarkable memory reduction capabilities of our MoE-SVD when applied to the Phi-3.5-MoE and Mixtral-8×7B models. As the compression ratio escalates, a substantial decrease in both the total memory and weight memory requirements is observed, closely aligning with the applied compression levels. For Phi-3.5-MoE model, 60% compression results in a weight memory reduction to 67.78 GB, a mere 43.45% of the original 155.99 GB. Similarly, Mixtral-8×7B exhibits a weight memory reduction to 70.31 GB at a 60% compression ratio, corresponding to 40.41% of its original 173.98 GB footprint. While a small portion of additional memory is required for auxiliary components, overall memory footprint reduction remains tightly coupled with our compression ratio. This significant memory reduction is particularly important for memory-limited devices, where every bit of memory counts. By reducing the memory requirements of these models, our MoE-SVD enables MoE LLMs to be deployed on a wider range of devices, making them more accessible and practical for real-world applications.

### 4.3 ABLATION STUDY

**Ablation of Selective Decomposition.** Table 3 delves into the performance of different selective decomposition methods. our non-uniform decomposition metric outperforms both uniform SVD and the OWL-based non-uniform SVD method for compressing MoE. In addition, our matrix sharing and trimming can further reduce the parameter redundancy allowing us to retain more sensitive expert layers, which leads to significant performance gains based on our selective decomposition.

**Varying Numbers for Low-rank Matrix Trimming.** Table 4 provides insights into the impact of matrix trimming on compressed MoE's performance. The results show a general trend of improved perplexity as the number of trimmed matrices increases. This is largely attributable to the reduction of experts, resulting in more stability in MoE LLMs. Based on this, we trim most of the U matrices for achieving the best balance between model size reduction and performance retention.

**Impact of Calibration Data.** Table 5 examines the impact of different calibration data sources and results indicate that the choice between WikiText-2 and C4 has minimal impact on the overall performance across various tasks. Figure 5 explores the effects of varying the number of calibration samples and random seed. Results indicate that increasing the number of data samples generally leads to a decrease in perplexity, suggesting improved performance with more samples. Additionally, the choice of random seed shows minimal effect on perplexity across both datasets, demonstrating that our MoE-SVD is relatively robust to sampling variability.

Table 7: Performance of our MoE-SVD with LoRA fine-tuning on Mixtral-8×7B and Phi-3.5-MoE.

| Method | WikiText-2↓ | PTB↓ | C4↓ | Openb. | ARC_e | WinoG. | HellaS. | ARC_c | PIQA | MathQA | Average↑ |
|---|---|---|---|---|---|---|---|---|---|---|---|
| Mixtral-8×7B Original | 3.98 | 12.99 | 6.78 | 0.36 | 0.84 | 0.76 | 0.65 | 0.57 | 0.82 | 0.43 | 0.63 |
| MoE-SVD (20%) | 5.94 | 19.42 | 8.98 | 0.28 | 0.75 | 0.69 | 0.55 | 0.45 | 0.78 | 0.36 | 0.55 |
| MoE-SVD (20%) +LoRA | 5.51 | 14.77 | 8.46 | 0.31 | 0.77 | 0.73 | 0.59 | 0.47 | 0.80 | 0.37 | 0.58 |
| MoE-SVD (50%) | 12.37 | 42.93 | 16.18 | 0.20 | 0.57 | 0.57 | 0.40 | 0.29 | 0.68 | 0.25 | 0.42 |
| MoE-SVD (50%) +LoRA | 8.73 | 23.36 | 12.13 | 0.25 | 0.67 | 0.64 | 0.50 | 0.37 | 0.73 | 0.28 | 0.49 |
| Phi-3.5-MoE Original | 3.48 | 8.43 | 8.22 | 0.40 | 0.77 | 0.76 | 0.68 | 0.56 | 0.79 | 0.38 | 0.62 |
| MoE-SVD (20%) | 4.77 | 12.12 | 9.56 | 0.39 | 0.77 | 0.69 | 0.59 | 0.53 | 0.74 | 0.35 | 0.58 |
| MoE-SVD (20%)+LoRA | 5.10 | 10.52 | 9.34 | 0.38 | 0.82 | 0.75 | 0.63 | 0.54 | 0.77 | 0.34 | 0.61 |
| MoE-SVD (50%) | 8.10 | 21.44 | 18.47 | 0.27 | 0.68 | 0.64 | 0.44 | 0.41 | 0.69 | 0.23 | 0.48 |
| MoE-SVD (50%) +LoRA | 7.90 | 15.74 | 13.20 | 0.27 | 0.74 | 0.68 | 0.47 | 0.42 | 0.73 | 0.27 | 0.51 |

Table 8: Perplexity (↓) of Mixtral 8x7B and Phi-3.5-MoE compressed with GPTQ and MoE-SVD with GPTQ on WikiText-2.

| Mixtral 8x7B | GPTQ (4bit) | GPTQ (3bit) | MoE-SVD (4bit) | MoE-SVD (3bit) | Phi-3.5-MoE | GPTQ (4bit) | GPTQ (3bit) | MoE-SVD (4bit) | MoE-SVD (3bit) |
|---|---|---|---|---|---|---|---|---|---|
| Memory | 44.5 | 33.4 | 35.6 | 26.7 | Memory | 39.0 | 29.3 | 27.3 | 20.5 |
| Perplexity | 4.35 | 6.22 | 6.93 | 11.53 | Perplexity | 4.59 | 6.71 | 6.64 | 10.28 |

**Generalizability of Across Diverse MoE Architectures.** To demonstrate the broad applicability of MoE-SVD, we conduct experiments on two distinct MoE models, DeepSeek-MoE-16B and Mixtral-8×22B in Table 6. Compressed MoE LLMs exhibit competitive performance on these datasets, with MoE-SVD achieving 0.42 average accuracy of on DeepSeek-MoE-16B and 0.57 on Mixtral-8×22B. These results reveal that our MoE-SVD can maintain a substantial portion of their original capabilities across various reasoning datasets.

**Improving MoE-SVD via LoRA Fine-Tuning.** Our MoE-SVD is training-free and can be further enhanced with additional fine-tuning. Table 7 confirms that the addition of LoRA fine-tuning on MoE-SVD shows some improvements, particularly at higher compression ratios. The Phi-3.5-MoE model appears to be more resilient to compression, maintaining better performance metrics compared to Mixtral-8×7B at equivalent compression ratios. These findings highlight the potential of combining our Moe-SVD with fine-tuning methods to mitigate performance losses in compressed models.

**Expanding MoE-SVD via Quantization.** The results in Table 8 demonstrate the results of combining MoE-SVD with GPTQ (Frantar et al., 2022) to achieve significant memory savings. Comparing 4-bit and 3-bit quantization levels, our MoE-SVD (4-bit) proves to be on par with direct 3-bit quantization (*e.g.*, GPTQ (3bit)) in terms of both memory efficiency and performance. These findings underscore the effectiveness of the quantization method when combined with MoE-SVD, showcasing its potential for creating memory-efficient models without compromising performance quality.

## 5 CONCLUSION

In this paper, we introduce MoE-SVD, a novel SVD-based compression framework tailored for MoE LLMs, effectively streamlining model parameters, computational expenses, and memory usage while upholding performance. To combat decomposition collapse stemming from matrix redundancy, we propose innovative solutions, including the selective decomposition strategy and a low-rank matrix sharing and trimming mechanism. The former utilizes a sensitivity metric for automated identification of decomposable layers, while the latter harmonizes parameter efficiency and expert specialization through V-matrix sharing and U-matrix trimming. Our extensive assessments on Mixtral-8×7B and Phi-3.5-MoE models showcase the method's superiority over existing compression techniques in preserving model capabilities across diverse tasks. These promising results, encompassing preserved performance, accelerated inference speed, and substantial memory reduction, position MoE-SVD as a significant stride forward in making MoE LLMs more accessible and efficient for real-world applications, paving the way for widespread adoption and deployment of these powerful models.

**Limitations:** To avoid confusion, we do not show results of combining our MoE-SVD with pruning method. In essence, our MoE-SVD is new technology and orthogonal to previous pruning-based approaches. In future work, we will strive to extend MoE-SVD with weight sparsity and pruning methods to achieve more extreme compression.

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

APPENDIX

Our appendix provides additional information and in-depth analysis to supplement the main content of the paper on MoE-SVD. It is organized into two main sections: further discussions and implementation details. The discussion section covers innovation, advantages and implications, disadvantages, and social implications of our proposed method. The implementation details section includes an algorithm table and specific implementation considerations.

## A    FURTHER DISCUSSIONS

### A.1: Ethics Statement

We focus solely on developing efficient techniques for Large Language Models (LLMs), utilizing publicly available datasets and models. Our research is not designed to address human ethics or privacy concerns directly. Instead, we concentrate on improving the computational efficiency and deployment capabilities of existing MoE LLMs, which may indirectly contribute to broader accessibility and utilization of these powerful models.

### A.2: Reproducibility

We affirm the solid reproducibility of our results and provide specific code implementations in the appendix. Our main experiments represent average outcomes from multiple repetitions, ensuring reliability. MoE LLMs, being very large models, exhibit relatively small variances in experimental results and evaluations. To further demonstrate the robustness and repeatability of our method, we present detailed results for different initial seeds, showcasing consistent performance across various conditions.

### A.3: Summary of Innovations

(1) We introduce MoE-SVD, the first SVD-based structured compression method specifically designed for MoE LLMs, addressing unique challenges such as decomposition sensitivity and expert redundancy. (2) Our selective decomposition strategy employs a novel sensitivity metric derived from matrix singular values and activation statistics, enabling adaptive compression across expert layers. (3) We develop low-rank matrix sharing and trimming techniques, including V-matrix sharing across experts and U-matrix trimming, significantly reducing parameters while maintaining expert diversity and model performance.

### A.4: Performance Gains

As the first SVD method developed for MoE LLMs, our approach demonstrates significant advantages in both performance and efficiency. (1) Our performance gains compared to other SVD methods are substantial, particularly at higher compression ratios. (2) Our main results are achieved without additional training, with potential for further improvement through fine-tuning. (3) We offer notable improvements in inference speed and memory optimization, crucial for practical deployment. (4) We maintain good performance even at very high compression ratios, a feat difficult for other compression methods to achieve.

### A.5: Comparison to BERT-based Compression Methods

(1) While low-rank decomposition methods exist for BERT-based MoE models (Li et al., 2024b), these are not applicable to MoE LLMs due to significant differences in model scale and architecture. We consider compression of MoE LLMs a distinct field, separate from BERT-based MoE compression. (2) Other LLM compression methods (Sun et al., 2023) also do not consider previous BERT-based compression techniques as direct competitors, recognizing the unique challenges posed by large-scale models.

### A.6: Comparison to Pruning-based Compression Methods

Our MoE-SVD approach and pruning-based methods (Frantar & Alistarh, 2023; Sun et al., 2023) are fully orthogonal, addressing different aspects of model compression. While pruning focuses on removing less important components, our method restructures the model through decomposition and sharing, offering complementary benefits. This orthogonality suggests potential for future research combining both approaches for even more efficient MoE LLM compression.

## B MORE RESULTS

### B.1: More Results on Per-layer Decomposition and Compression Ratios

Our extended experimental investigations provide deeper insights into the efficacy and behavior of the MoE-SVD compression technique. Figure 7 presents a detailed distribution of sensitivity scores across layers for both Mixtral-8×7B and Phi-3.5-MoE models. This analysis elucidates the varying impact of compression on different layers within the network architecture. Furthermore, we conduct an in-depth examination of perplexity results for each decomposed block of Mixtral-8×7B at 20% compression,

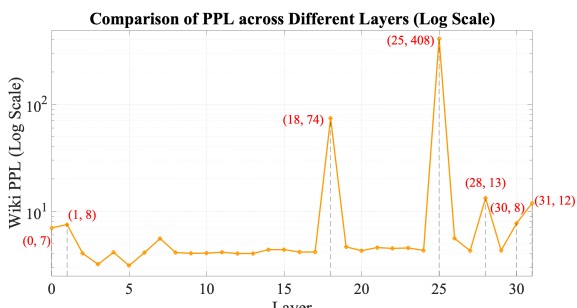

Figure 6: Perplexity of Mixtral-8×7B via 20% per-layer SVD decomposition on WikiText-2.

as illustrated in Figure 6. These results offer valuable insights into the relationship between compression ratios and model performance. To further optimize our layer selection process, we develop and implement a sophisticated heatmap-based approach, visualized in Figure 9. This method provides a more intuitive and data-driven way to identify layers most suitable for compression, enhancing the overall efficiency and effectiveness of our MoE-SVD technique.

### B.2: More Comparison with Structured Compression Methods

In Table 9, we compare our MoE-SVD against several structured compression methods (Frantar & Alistarh, 2023; Sun et al., 2023; Li et al., 2023; Lee et al., 2024) applied to Mixtral-8×7B, as well as methods specifically targeting expert layer compression (Li et al., 2024b; He et al., 2024). Our MoE-SVD achieves a runtime speedup of 1.2× while maintaining performance across various benchmarks. Specifically, MoE-SVD records lower perplexities on language modeling tasks such as WikiText-2 (4.44) and PTB (15.21) compared to Wanda (4.72 and 18.8) and SparseGPT (4.61 and 21.11). On downstream tasks, our method attains the highest average score of 0.58, outperforming Unified-MoE-Compress (He et al., 2024)'s 0.54 and significantly surpassing LoSparse (Li et al., 2023) and MC-SMoE (Li et al., 2024b), which exhibit substantial performance drops. These results highlight that MoE-SVD not only accelerates inference but also preserves or improves accuracy relative to other methods.

### B.3: More Results with 512 Calibrated Samples, Real-time and Significance test

In Tables 10, 11, and 12, we present the experimental results of our MoE-SVD method applied to Mixtral-8×7B and Phi-3.5-MoE models at 20% ratios. Our approach achieves substantial reductions in model size and computational overhead while maintaining competitive performance. Specifically, MoE-SVD reduces the model size of Mixtral-8×7B from 46.7B to 37.1B and improves runtime throughput from 87.73 to 104.66 Tokens/Sec. In terms of PPLs, using 512 calibrated samples results in lower perplexities compared to 256 samples. For example, the PPL on WikiText-2 decreases from 5.94 to 4.44. Additionally, combining MoE-SVD with LoRA fine-tuning further enhances performance, raising the average score on downstream tasks from 0.58 to 0.60 for Mixtral-8×7B. The significance tests in Table 12 indicate that these improvements are statistically meaningful across multiple runs.

### B.4: More Results on Qwen Model

Table 13 presents the performance of the Qwen2-57B-A14B model compressed using our MoE-SVD method under a 20% compression ratio. The compressed model achieves a throughput of 53.16 Tokens/sec, representing a 1.25× increase over the original model's 42.7 Tokens/sec. While there is a moderate increase in PPL, WikiText-2 PPL rises from 4.32 to 5.41 and the average score on downstream tasks decreases only slightly from 0.58 to 0.56 with LoRA fine-tuning after MoE-SVD. These results show that MoE-SVD effectively enhances runtime efficiency with minimal impact on model accuracy and PPL.

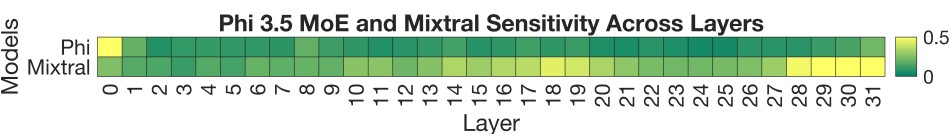

Figure 7: Sensitivity scores of Mixtral-8×7B and Phi-3.5-MoE accrossing layers

## C PSEUDOCODE

In our experimental implementation, we present a detailed algorithmic procedure for compressing MoE-based large language models using the proposed MoE-SVD method. Algorithm 4 outlines the main steps of this approach. The process begins by collecting scaling matrices through forward hooks during inference, as shown in Algorithm 1 (Step 1). This step is crucial for capturing activation patterns and computing the sensitivity metric for each expert. Subsequently, we perform singular value decomposition (SVD) on the scaled weight matrices, followed by truncation for effective compression, as detailed in Algorithm 2 (Step 2). Our method introduces a V-matrix sharing mechanism, where the most frequently used V-matrix is selected and shared among all experts, as described in Algorithm 3 (Step 3). Additionally, we employ U-matrix trimming by retaining the top-$k$ U-matrices based on expert sampling frequencies to refine the expert functions (Step 4). To ensure numerical stability, we apply the adjustment function provided in Algorithm 5, which modifies matrices to be positive definite when necessary. This comprehensive approach enables significant model compression while maintaining performance, effectively addressing the need for efficient large-scale language models.

## D IMPLEMENTATION DETAILS

MoE-SVD optimizes MoE models by selectively decomposing less critical experts to reduce computational complexity while maintaining performance. It consists of two main phases: computing a sensitivity metric $S_L$ for each expert layer during calibration data inference, and decomposing experts.

Table 9: Performance of Mixtral-8×7B compressed by MoE-SVD under 20% compression ratios.

| Method | Runtime speedup | WikiText-2↓ | PTB↓ | C4↓ | Openb. | ARC_e | WinoG. | HellaS. | ARC_c | PIQA | MathQA | Average↑ |
|---|---|---|---|---|---|---|---|---|---|---|---|---|
| Wanda (2:4) | 1.04× | 4.72 | 18.8 | 8.43 | 0.32 | 0.76 | 0.72 | 0.55 | 0.47 | 0.79 | 0.36 | 0.57 |
| SparseGPT (2:4) | 1.06× | 4.61 | 21.11 | 8.19 | 0.3 | 0.77 | 0.74 | 0.56 | 0.45 | 0.77 | 0.35 | 0.56 |
| LoSparse | 1.06× | 953.51 | 805.16 | 1273.12 | 0.2 | 0.27 | 0.49 | 0.28 | 0.26 | 0.53 | 0.2 | 0.32 |
| MC-SMoE | 1.09× | 1341.36 | 1316.52 | 1478.13 | 0.26 | 0.28 | 0.51 | 0.29 | 0.25 | 0.54 | 0.19 | 0.33 |
| Unified-MoE | 1.13× | 6.12 | 14.67 | 11.61 | 0.3 | 0.73 | 0.7 | 0.54 | 0.46 | 0.73 | 0.33 | 0.54 |
| MoE-SVD | 1.2× | 4.44 | 15.21 | 8.32 | 0.32 | 0.78 | 0.73 | 0.57 | 0.48 | 0.79 | 0.37 | 0.58 |

Table 10: Metrics (model size, TFLOPs, runtime) of MoE-SVD. Runtime denotes runtime throughput (Tokens/sec) on a single H800 GPU.

| Mixtral-8×7B | Dense | 20% | 30% | 40% | 50% | 60% |
|---|---|---|---|---|---|---|
| Model-size | 46.7B | 37.1B | 32.2B | 28.3B | 23B | 17.6B |
| TFLOPs | 5.27E+14 | 4.92E+14 | 4.40E+14 | 4.26E+14 | 3.97E+14 | 3.67E+14 |
| Runtime | 87.73 | 104.66 | 106.03 | 108.83 | 123.88 | 156.1 |

| Phi-3.5-MoE | Dense | 20% | 30% | 40% | 50% | 60% |
|---|---|---|---|---|---|---|
| Model-size | 41.9B | 33.2B | 29B | 25.2B | 20.6B | 16.4B |
| TFLOPs | 2.72E+14 | 2.46E+14 | 2.27E+14 | 2.00E+14 | 1.82E+14 | 1.71E+14 |
| Runtime | 98.2 | 108.63 | 114.8 | 124.79 | 137.17 | 148.7 |
| DeepSeekMoE | Dense | 20% | 30% | 40% | 50% | 60% |

| Model-size | 6.4B | 13.2B | 11.4B | 9.7B | 8B | 6.4B |
|---|---|---|---|---|---|---|
| TFLOPs | 1.10E+14 | 1.02E+14 | 9.88E+13 | 9.29E+13 | 9.25E+13 | 8.82E+13 |
| Runtime | 52.53 | 62.79 | 94.71 | 118.93 | 119.81 | 128.71 |

**D.1: Calculation of Sensitivity Score** During calibration data inference, both the sensitivity metric $S_L = \sum_{i=1}^{N} f_i \cdot r_i \cdot a_i$ and the activation matrices for each expert $i$ are collected. The sensitivity

Table 11: Performance of Mixtral-8×7B and Phi-3.5-MoE compressed by MoE-SVD under 20% compression ratios.

| Mixtral-8×7B | WikiText-2↓ | PTB↓ | C4↓ | Openb. | ARC_e | WinoG. | HellaS. | ARC_c | PIQA | MathQA | Average↑ |
|---|---|---|---|---|---|---|---|---|---|---|---|
| Original | 3.98 | 12.99 | 6.78 | 0.36 | 0.84 | 0.76 | 0.65 | 0.57 | 0.82 | 0.43 | 0.63 |
| MoE-SVD (256) | 5.94 | 19.42 | 8.98 | 0.28 | 0.75 | 0.69 | 0.55 | 0.45 | 0.78 | 0.36 | 0.55 |
| MoE-SVD (512) | 4.44 | 15.21 | 8.32 | 0.32 | 0.78 | 0.73 | 0.57 | 0.48 | 0.79 | 0.37 | 0.58 |
| MoE-SVD (512)+LoRA | 4.31 | 14.94 | 7.82 | 0.33 | 0.8 | 0.73 | 0.61 | 0.55 | 0.81 | 0.38 | 0.6 |
| Phi-3.5-MoE | WikiText-2↓ | PTB↓ | C4↓ | Openb. | ARC_e | WinoG. | HellaS. | ARC_c | PIQA | MathQA | Average↑ |
| Original | 3.48 | 8.43 | 8.22 | 0.4 | 0.77 | 0.76 | 0.68 | 0.56 | 0.79 | 0.38 | 0.62 |
| MoE-SVD (256) | 4.77 | 12.12 | 9.56 | 0.39 | 0.77 | 0.69 | 0.59 | 0.53 | 0.74 | 0.35 | 0.58 |
| MoE-SVD (512) | 4.26 | 11.41 | 9.53 | 0.38 | 0.76 | 0.72 | 0.63 | 0.53 | 0.77 | 0.35 | 0.59 |
| MoE-SVD (512) +LoRA | 4.29 | 10.99 | 8.81 | 0.39 | 0.81 | 0.74 | 0.65 | 0.54 | 0.79 | 0.36 | 0.61 |

Table 12: Significance test for three repeated experiments of Mixtral-8×7B and Phi-3.5-MoE compressed by MoE-SVD under 20% compression ratios.

| Mixtral-8×7B | Openb. | ARC_e | WinoG. | HellaS. | ARC_c | PIQA | MathQA |
|---|---|---|---|---|---|---|---|
| MoE-SVD (512) | 0.32±0.0199 | 0.78±0.0088 | 0.73±0.0129 | 0.57±0.0050 | 0.48±0.0146 | 0.79±0.0096 | 0.37±0.0087 |
| MoE-SVD (512) +LoRA | 0.33±0.0205 | 0.80±0.0086 | 0.73±0.0128 | 0.61±0.0049 | 0.55±0.0145 | 0.81±0.0095 | 0.38±0.0084 |
| Phi-3.5-MoE | Openb. | ARC_e | WinoG. | HellaS. | ARC_c | PIQA | MathQA |
| MoE-SVD (512) | 0.38±0.0215 | 0.76±0.0090 | 0.72±0.0127 | 0.63±0.0049 | 0.53±0.0146 | 0.77±0.0099 | 0.35±0.0081 |
| MoE-SVD (512) +LoRA | 0.39±0.0213 | 0.81±0.0080 | 0.74±0.0123 | 0.65±0.0041 | 0.54±0.0142 | 0.79±0.0095 | 0.36±0.0084 |

metric integrates utilization frequency $f_i$, principal Rank $r_i$, and activation outliers $a_i$, where each component is described in detail below:

Sampling Frequency ($f_i$): The variable $f_i$ represents the utilization frequency of the $i$-th expert, quantifying how often this expert is selected by the router during inference. It is calculated over a calibration dataset $\mathcal{X}$ as:

$$f_i = \frac{\sum_{x \in \mathcal{X}} \mathbb{I}[i \in \text{TopK}(G(x), k)]}{|\mathcal{X}|}, \tag{9}$$

where $G(x)$ is the output of the gating network for input $x$, $\text{TopK}(G(x), k)$ returns the indices of the top $k$ selected experts, $\mathbb{I}[\cdot]$ is the indicator function, and $|\mathcal{X}|$ denotes the total number of samples in the dataset. This metric reflects the relative importance of each expert based on its selection frequency.

Principal Rank ($r_i$): The variable $r_i$ denotes the principal rank of the $i$-th expert, which is the number of dominant singular values in the diagonal matrix $\Sigma_i$ obtained from the SVD of the expert's weight matrix $W_i$. $r_i$ is defined as the number of singular values in $\Sigma_i$ that exceed a given threshold, effectively capturing the dimensionality of the weight matrix's significant components. This rank reflects the structural complexity of the expert's weight representation, with higher values of $r_i$ indicating more complex and information-rich weights.

Activation Outliers ($a_i$): The variable $a_i$ measures the proportion of activations in the $i$-th expert that exceed a certain threshold relative to the mean absolute activation value. For a set of activations $A_i$ in the $i$-th expert, $a_i$ is computed as:

$$a_i = \frac{\sum_{a \in A_i} \mathbb{I}(|a| > \tau \cdot \text{Mean}(|A_i|))}{|A_i|}, \tag{10}$$

where $|A_i|$ denotes the total number of activations for the $i$-th expert, $\text{Mean}(|A_i|)$ is the mean absolute value of these activations, and $\tau$ is a user-defined threshold. This metric highlights the presence of outlier activations indicative of the expert's contribution to the model's capacity. The overall sensitivity metric $S_L$ aggregates these factors across all $N$ experts in a layer, providing a comprehensive measure of the layer's importance.

Table 13: Performance of Qwen2-57B-A14B compressed by MoE-SVD under 20% compression ratios.

| Method | Throughput (Tokens/sec) | WikiText-2↓ | PTB↓ | C4↓ | Openb. | ARC_e | WinoG. | HellaS. | ARC_c | PIQA | MathQA | Average↑ |
|---|---|---|---|---|---|---|---|---|---|---|---|---|
| Original | 42.7 | 4.32 | 11.66 | 9.23 | 0.33 | 0.75 | 0.74 | 0.63 | 0.47 | 0.8 | 0.39 | 0.58 |
| Qwen2-57B-A14B | 53.16 (1.25×) | 6.52 | 14.61 | 13.64 | 0.29 | 0.71 | 0.69 | 0.58 | 0.42 | 0.74 | 0.33 | 0.53 |
| Qwen2-57B-A14B + Lora | 53.16 (1.25×) | 5.41 | 13.26 | 11.63 | 0.3 | 0.74 | 0.73 | 0.61 | 0.45 | 0.78 | 0.35 | 0.56 |

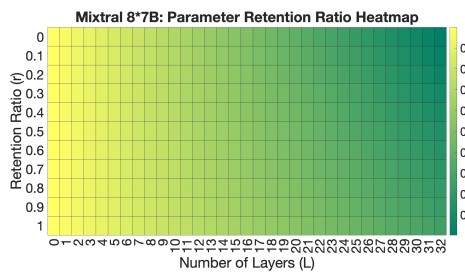 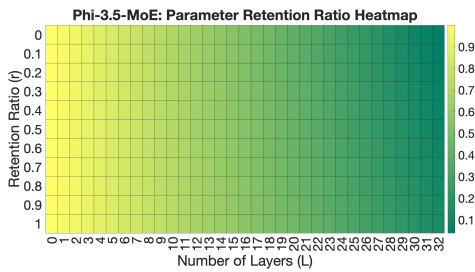

Figure 8: Retained parameters calculation for Mixtral-8×22B.

Figure 9: Retained parameters calculation for Phi-3.5-MoE.

### D.2: Decomposition Process of Expert Matrix

Following ASVD (Yuan et al., 2023) and SVD-LLM (Wang et al., 2024), our framework employs an activation-weighted SVD that enhances the vanilla SVD by incorporating activation statistics to improve decomposition accuracy. With activation matrix $X$ and original weight $W_{\text{original}}$, we compute the activation-weighted matrix by scaling the original weight matrix based on the activation statistics:

$$W_{\text{aw}} = W_{\text{original}} \cdot S, \tag{11}$$

where $W_{\text{aw}} \in \mathbb{R}^{m \times n}$ represents the activation-weighted matrix and matrix $S$ is obtained through cholesky decomposition of activation gram matrix $\mathbf{X}\mathbf{X}^T$. We then perform SVD on $W_{\text{aw}}$ and final compressed weight matrix is obtained by truncating the smallest singular values:

$$W_{\text{aw}} = U \cdot \text{Trunc}(\Sigma) \cdot V^T \cdot S^{-1}, \tag{12}$$

This activation-weighted approach effectively mitigates reconstruction loss from outliers during matrix decomposition while maintaining the essential characteristics of the original weight distribution.

### D.4: Model-Specific Configurations

In our experimental realization, we develop tailored post-decomposition strategies for various large language models, each with unique architectures. For Mixtral-8×7B, which employs 8 experts per layer, we implement a novel approach of sharing V components (v1, v2, v3) from the most frequently activated expert while retaining U components (u1, u2, u3) from the top two most frequent experts. We extend this methodology to Phi-3.5-MoE, featuring 16 experts per layer, by broadening the retention scope. In this instance, we share V components from the most frequently selected expert and preserve U components from the top four most frequent experts. For the more complex Deepseek 16B, which utilizes 64 experts per block, we innovate further by partitioning the experts into 8 distinct groups. Within each group, we share the most frequent V component and strategically trim the 6 lowest frequency U components. This carefully crafted selective retention and sharing approach enables us to maintain model performance while achieving substantial reductions in both parameter count and computational requirements.

### D.5: Computational Efficiency

We rigorously assess the computational efficiency of our LLM compression techniques, recognizing its paramount importance for practical deployment scenarios. Our MoE-SVD compression methodology comprises two distinct phases: activation data collection and SVD decomposition with expert trimming. Through extensive experimentation on the Phi 3.5 MoE model, we meticulously quantify time requirements for various layer counts. Our findings reveal that single-layer processing consumes 266 seconds for activation collection and 107 seconds for SVD and trimming. These durations exhibit a non-linear increase, reaching 489 and 209 seconds for two layers, and 689 and 320 seconds for three layers, respectively. In a comprehensive 20% layer compression test, we observe a total time requirement of 44 minutes, with 30 minutes allocated to data collection and 14 minutes to SVD and trimming. These results provide crucial insights into the scalability and efficiency of our approach across different model configurations.

### D.6: Scalability Analysis

Our in-depth scalability analysis unveils intriguing patterns in the computational behavior of our compression technique. The activation collection phase demonstrates near-linear time growth with respect to layer count, indicating limited parallelization potential due to the inherent sequential nature of data propagation. However, we emphasize that this collection process is a one-time operation per model, with the resulting data being storable and reusable for various compression ratios. In contrast, the SVD and trimming phase exhibits promising sub-linear scaling, suggesting enhanced opportunities for parallelization. While our current implementation relies on sequential Python loops, we identify significant potential for efficiency improvements through parallel processing of experts across layers. This aligns seamlessly with the independent operation of experts in different layers of MoE models, indicating promising scalability prospects for MoE-SVD, particularly in the computationally intensive SVD and trimming phase when applied to large-scale models.

### D.7: Potential Parallelization Strategies

To further optimize our approach, we explore a range of potential parallelization strategies. We consider leveraging Python's multiprocessing modules and GPU acceleration frameworks such as PyTorch or TensorFlow to exploit parallel computing capabilities. For models exceeding 100B parameters, we propose the utilization of distributed computing frameworks like Dask or Ray to efficiently scale computation across multiple machines. We hypothesize that this approach could potentially reduce SVD phase time complexity from $O(mn^2)$ to near-linear relative to processor count, with the potential to scale with the maximum expert count per layer rather than the total layer count. However, we acknowledge that the effectiveness of these strategies may vary based on available computational resources, inter-process communication overhead, and the challenges of expert load balancing in distributed environments. In addressing the research challenges associated with our proposed parallelization strategies, we encounter several non-trivial technical hurdles. These include the need to fundamentally redesign algorithms for efficient concurrent processing, develop robust mechanisms for managing complex data dependencies, and optimize resource utilization across heterogeneous computing environments. We emphasize the critical importance of conducting comprehensive empirical studies to quantify potential performance improvements across a diverse range of model sizes and hardware configurations. While we anticipate that parallel processing may significantly enhance MoE-SVD's scalability for large language models, we maintain a cautious stance regarding its effectiveness when combined with our existing matrix sharing and trimming optimizations. We assert that rigorous experimentation and thorough analysis are essential to verify these potential benefits and to fully understand the implications of our proposed parallelization strategies in real-world, large-scale language model compression scenarios.

**Algorithm 1:** PyTorch code for sensitivity metric of MoE-SVD.

```python
import torch
import torch.nn as nn

def compute_layer_sensitivity(experts_weights, activations, gating_outputs,
    calibration_data, top_k=2, tau=2.0):
    """
    Compute layer-wise sensitivity metric S_L for MoE compression

    Args:
        experts_weights (list of torch.Tensor): Weight matrices for each expert
        activations (list of torch.Tensor): Activation values for each expert
        gating_outputs (torch.Tensor): Router outputs for calibration data
        calibration_data (torch.Tensor): Calibration dataset
        top_k (int): Number of experts to select per token
        tau (float): Threshold for activation outliers

    Returns:
        float: Layer sensitivity score S_L
    """
    num_experts = len(experts_weights)
    device = experts_weights[0].device

    # Compute sampling frequency (f_i)
    top_k_indices = torch.topk(gating_outputs, top_k, dim=-1).indices
    expert_counts = torch.zeros(num_experts, device=device)
    for indices in top_k_indices:
        expert_counts[indices] += 1
    f_i = expert_counts / len(calibration_data)

    # Compute principal rank (r_i) using SVD
    r_i = torch.zeros(num_experts, device=device)
    for i, weight in enumerate(experts_weights):
        U, S, V = torch.linalg.svd(weight)
        # Count singular values above threshold
        threshold = torch.max(S) * 1e-2 # Example threshold
        r_i[i] = torch.sum(S > threshold)

    # Compute activation outliers (a_i)
    a_i = torch.zeros(num_experts, device=device)
    for i, activation in enumerate(activations):
        mean_abs_act = torch.mean(torch.abs(activation))
        outliers = torch.sum(torch.abs(activation) > tau * mean_abs_act)
        a_i[i] = outliers / activation.numel()

    # Compute final sensitivity metric S_L
    S_L = torch.sum(f_i * r_i * a_i)

    return S_L
```

**Algorithm 2:** PyTorch code for SVD Expert Decomposition of MoE-SVD.

```python
class MoESVDCompression:
    def __init__(self, truncate_k=None, top_k_experts=2):
        """
        Initialize Activation-Weighted SVD with Matrix Sharing and Trimming

        Args:
            truncate_k (int): Number of singular values to keep
            top_k_experts (int): Number of top experts to select for U-matrix
                trimming
        """
        self.truncate_k = truncate_k
        self.top_k_experts = top_k_experts

    def compute_activation_weights(self, X):
        """
        Compute activation-weighted scaling matrix S using Cholesky decomposition

        Args:
            X (torch.Tensor): Activation matrix [batch_size,feature_dim]
            torch.mm(X, X.t()) is Cumulative activation matrix, representing the sum
                of processed activation data.
        """
        # Compute Gram matrix
        gram = torch.mm(X, X.t())

        # Cholesky decomposition
        S = torch.linalg.cholesky(gram)
        return S

    def decompose_expert(self, W_original, X):
        """
        Perform activation-weighted SVD on single expert

        Args:
            W_original (torch.Tensor): Original weight matrix
            X (torch.Tensor): Activation matrix
        """
        # Compute activation-weighted matrix
        S = self.compute_activation_weights(X)
        W_aw = torch.mm(W_original, S)

        # Perform SVD
        U, sigma, V = torch.linalg.svd(W_aw, full_matrices=False)

        # Truncate if specified
        if self.truncate_k is not None:
            U = U[:, :self.truncate_k]
            sigma = sigma[:self.truncate_k]
            V = V[:self.truncate_k, :]

        return U, torch.diag(sigma), V, S
```

**Algorithm 3:** PyTorch code for Matrix Sharing & Trimming of MoE-SVD

```python
class MoESVDCompression:
    def __init__(self, truncate_k=None, top_k_experts=2):
        """
        Initialize Activation-Weighted SVD with Matrix Sharing and Trimming

        Args:
            truncate_k (int): Number of singular values to keep
            top_k_experts (int): Number of top experts to select for U-matrix
                trimming
        """
        self.truncate_k = truncate_k
        self.top_k_experts = top_k_experts
    def __init__(self, truncate_k=None, top_k_experts=2):
        """
        Initialize Activation-Weighted SVD with Matrix Sharing and Trimming

        Args:
            truncate_k (int): Number of singular values to keep
            top_k_experts (int): Number of top experts to select for U-matrix
                trimming
        """
        self.truncate_k = truncate_k
        self.top_k_experts = top_k_experts

    def compress_moe(self, expert_weights, activations, routing_frequencies):
        """
        Compress MoE using V-matrix sharing and U-matrix trimming

        Args:
            expert_weights (list): List of expert weight matrices
            activations (list): List of activation matrices for each expert
            routing_frequencies (torch.Tensor): Expert selection frequencies
        """
        num_experts = len(expert_weights)
        compressed_experts = []

        # Decompose all experts
        decomposed = []
        for i in range(num_experts):
            U, Sigma, V, S = self.decompose_expert(expert_weights[i], activations[i])
            decomposed.append((U, Sigma, V, S))

        # Select shared V-matrix based on highest routing frequency
        max_freq_idx = torch.argmax(routing_frequencies)
        V_shared = decomposed[max_freq_idx][2]

        # Sort experts by routing frequency for U-matrix trimming
        sorted_indices = torch.argsort(routing_frequencies, descending=True)

        # Perform U-matrix trimming and construct compressed experts
        for i in range(num_experts):
            # Find top-k U-matrices from more frequently used experts
            more_frequent = [j for j in sorted_indices if routing_frequencies[j] >
                routing_frequencies[i]]
            top_k_indices = more_frequent[:self.top_k_experts]

            if len(top_k_indices) < self.top_k_experts:
                # If not enough more frequent experts, use own U-matrix
                top_k_indices = top_k_indices + [i]

            # Combine selected U-matrices and corresponding Sigma matrices
            U_combined = torch.zeros_like(decomposed[i][0])
            Sigma_combined = torch.zeros_like(decomposed[i][1])

            for idx, expert_idx in enumerate(top_k_indices[:self.top_k_experts]):
                U_combined += decomposed[expert_idx][0]
                Sigma_combined += decomposed[expert_idx][1]

            # Reconstruct compressed expert
            W_compressed = torch.mm(torch.mm(U_combined, Sigma_combined),
                            torch.mm(V_shared, torch.inverse(decomposed[i][3])))
            compressed_experts.append(W_compressed)

        return compressed_experts
```

**Algorithm 4:** SVD-MOE: Expert Decomposition, V-Matrix Sharing, and U-Matrix Trimming

**Input:** model: Language model, data: Calibration data, device: Device, layers: Selected layers, config: Expert configuration

**Output:** model_compressed: Compressed model

$\triangleright$ **Step 1: Calibration Processing**

**for** *layer $\ell$ in layers* **do**
    **for** *expert $i$ in $\ell$* **do**
        $f_i \leftarrow \frac{i \in \text{TopK}(\text{G(x),K})}{|\mathcal{X}|}$;
                                         $\triangleright$ Update activation matrix
        $S_i \leftarrow S_i + AA^T$;
        $U, \Sigma, V^T \leftarrow \text{SVD}(expert_i)$;
        $r_i \leftarrow \text{Rank}(\Sigma)$;
                                         $\triangleright$ Update weight outlier
        $\overline{W}_i \leftarrow \text{mean}(expert_i.W)$;
        $a_i \leftarrow +\frac{\#\{|A| > \alpha \overline{W}_i\}}{A}$;
        $expert_i.S_L \leftarrow expert_i.S_L + f_i r_i a_i$;

$\triangleright$ **Step 2: Decompose Experts**

**for** *layer $\ell$ in layers* **do**
    **for** *expert $i$ in $\ell$* **do**
        **if** $expert_i.S_L < \tau$ **then**
            **if** *$S_i$ is not positive definite* **then**
                Adjust $S_i$
            $W_s \leftarrow expert_i.W \times S_i$;
            $U, \Sigma, V^T \leftarrow \text{SVD}(W_s)$;
            $\Sigma_{trunc} \leftarrow Truncate(\Sigma)$;
            $expert_i.U \leftarrow U\Sigma_{trunc}$;
            $expert_i.V \leftarrow V^T$;

                                  $\triangleright$ **Step 3: V-Matrix Sharing**
                                     $\triangleright$ Select shared V-matrix
    $V_s \leftarrow \arg\max_{V_i} f(V_i)$;
                                       $\triangleright$ Select top-2 U-matrices
    $\{U_{i,1}, U_{i,2}\} \leftarrow \text{TopK}(\{U_j\}, k = 2)$;
                        $\triangleright$ **Step 4: U-Matrix Trimming and Selection**
                                    $\triangleright$ Update expert function
    $E_i(x) \leftarrow (U_{i,1}\Sigma_{i,1} + U_{i,2}\Sigma_{i,2})V_s^T x$;

1296
1297
1298
1299
1300
1301
1302
1303
1304
1305
1306
1307
1308
1309
1310
1311
1312
1313
1314
1315
1316
1317
1318
1319
1320
1321
1322
1323
1324
1325
1326
1327
1328
1329
1330
1331
1332
1333
1334
1335
1336
1337
1338
1339
1340
1341
1342
1343
1344
1345
1346
1347
1348
1349

---

**Algorithm 5:** MakePositiveDefinite: Adjust Matrix to be Positive Definite

---

**Function** MakePositiveDefinite($M$, *tolerance, max_attempts*) **:**

    **Input:** $M$ - Input matrix; tolerance - Small value for adjustment; max_attempts - Maximum number of attempts

    **Output:** $M_{\text{pd}}$ - Positive definite matrix

    **Step 1: Symmetrize the Matrix**;

    $M_{\text{sym}} \leftarrow \frac{M+M^T}{2}$;

    // Ensure the matrix is symmetric

    **Step 2: Check Eigenvalues**;

    Compute eigenvalues $\lambda$ of $M_{\text{sym}}$;

    **if** *any* $\lambda_i < 0$ **then**

        $\lambda \leftarrow \lambda + |\min(\lambda_i)| + \text{tolerance}$;

        // Shift negative eigenvalues to positive

    **end**

    **Step 3: Reconstruct Positive Definite Matrix**;

    $M_{\text{pd}} = V \, \text{diag}(\lambda) \, V^T$;

    where $V$ are the eigenvectors of $M_{\text{sym}}$;

    **Step 4: Ensure Matrix is Symmetric**;

    $M_{\text{pd}} \leftarrow \frac{M_{\text{pd}}+M_{\text{pd}}^T}{2}$;

    **return** $M_{\text{pd}}$;

---

