# OpenReview forum: "Structured Mixture-of-Experts LLMs Compression  via Singular Value Decomposition"
_ICLR.cc/2025/Conference — Submitted to ICLR 2025_

### Official Review · Reviewer_nYc5 · 2024-10-31

**Soundness:** 3
**Presentation:** 3
**Contribution:** 2
**Rating:** 6
**Confidence:** 4

**Summary:**

This paper proposes a compression framework called MoE-SVD, which uses a low-rank matrix decomposition of experts and a selective decomposition strategy based on weight singular values and activation statistics to identify the factorable expert layer, while sharing the V-matrix and implementing top-k selection in the U-matrix to reduce the number of parameters while maintaining the diversity among experts. The experimental results show that MoE-SVD performs better than existing compression methods on multiple datasets in terms of performance and efficiency tradeoff.

**Strengths:**

1. The method seems technically sound and straightforward in principle.
2. Good written and easy to follow.

**Weaknesses:**

1. The accuracy degradation of the compressed model is too obvious. With a 20% compression ratio, the average accuracy is generally reduced by 5%-10%. Even with the LoRA fine-tuning, the compression model cannot be improved to lossless, which greatly limits the practicability of the method.
2. Once it comes to fine-tuning, the author needs to compare it with some fine-tuning compression methods, such as MC-SMoE [1], which also uses low-rank decomposition + sparse technique.
3. For matrix multiplication, Rank(AB) ≤ min(Rank(A), Rank(B)), which also means that for an MLP layer, the rank of its output y is often less than W and x. Maybe using AFM [2] instead of SVD here is better, i.e., perform SVD decomposition of MLP output y, and then merge UV into the MLP weight. According to paper [2], AFM seems to be consistently superior to SVD.
4. The authors only test the throughput in Mixtral-8x7B and Phi-3.5-MoE, lacking speed on DeepSeekMoE. This is a little strange, because some MoE LLMs (such as DeepSeekMoE and Qwen2-57B-A14B) have a very heavy share-expert, which is a key bottleneck in LLM inference. Can the author discuss the actual speed performance of MOE-SVD on this type of model?
5. For current LLM inference, to achieve higher throughput, tensor parallelism is generally adopted. However, low-rank decomposition will split one MLP layer into two layers, which may result in relatively large communication overhead when combined with tensor parallelism. At present, the throughput measurement has only been done on an H100 GPU. Can authors further discuss the tensor parallelism (like TP4) inference with  MoE-SVD in detail?

[1]: Merge, Then Compress: Demystify Efficient SMoE with Hints from Its Routing Policy, In ICLR2024.
[2]: Compressing transformers: features are low-rank, but weights are not! In AAAI 2023.

**Questions:**

Please see the weakness.

---

> ### Author Response · Authors · 2024-11-22
> **Response to Reviewer nYc5 (Part 1/5)**
>
> **Dear Reviewer nYc5,**
>
> Thanks for the valuable feedback. We have tried our best to address all concerns in the last few days. If the reviewer finds our response adequate, **we would really appreciate it if the reviewer considers raising the score.** Please see our responses below one by one：
>
> ------
>
> **Q1: About accuracy degradation.**
>
> **A1:**  (**1) Our MoE-SVD performance can continue improving with more calibration samples.** Our original setting of calibration sample size is 64 the same as that of SVD-LLM for general LLM. During rebuttal, we observed that existing MoE compression methods like MC-SMoE [4] employ larger calibration sample sizes. We have aligned the calibration samples (512) accordingly, and the updated results for MoE-SVD (512) are presented in the following Table. The results indicate that our MoE-SVD method, without fine-tuning, exhibits only ~3% reduction in performance at a 20% compression ratio respectively  on Phi3.5 MoE and Mixtral 8x7B, which is clearly superior to other compression methods. **Additionally, our MoE-SVD  with LoRA fine-tuning achieves performance levels close to that of the original dense MoE model.**
>
> **Table A: Performance of Mixtral-8×7B and Phi-3.5-MoE compressed by MoE-SVD with 512 calibration samples under 20% compression ratios.**
>
> | Mixtral-8×7B            | WikiText-2↓     | PTB↓      | C4↓      | Openb.     | ARC_e     | WinoG.     | HellaS.     | ARC_c     | PIQA     | MathQA     | Average↑     |
> | ----------------------- | --------------- | --------- | -------- | ---------- | --------- | ---------- | ----------- | --------- | -------- | ---------- | ------------ |
> | Original                | 3.98            | 12.99     | 6.78     | 0.36       | 0.84      | 0.76       | 0.65        | 0.57      | 0.82     | 0.43       | 0.63         |
> | MoE-SVD (256)           | 5.94            | 19.42     | 8.98     | 0.28       | 0.75      | 0.69       | 0.55        | 0.45      | 0.78     | 0.36       | 0.55         |
> | MoE-SVD (512)           | 4.44            | 15.21     | 8.32     | 0.32       | 0.78      | 0.73       | 0.57        | 0.48      | 0.79     | 0.37       | 0.58         |
> | **MoE-SVD (512) +LoRA** | **4.31**        | **14.94** | **7.82** | **0.33**   | **0.80**  | **0.73**   | **0.61**    | **0.55**  | **0.81** | **0.38**   | **0.60**     |
> | **Phi3.5 MoE**          | **WikiText-2↓** | **PTB↓**  | **C4↓**  | **Openb.** | **ARC_e** | **WinoG.** | **HellaS.** | **ARC_c** | **PIQA** | **MathQA** | **Average↑** |
> | Original                | 3.48            | 8.43      | 8.22     | 0.40       | 0.77      | 0.76       | 0.68        | 0.56      | 0.79     | 0.38       | 0.62         |
> | MoE-SVD (256)           | 4.77            | 12.12     | 9.56     | 0.39       | 0.77      | 0.69       | 0.59        | 0.53      | 0.74     | 0.35       | 0.58         |
> | MoE-SVD (512)           | 4.26            | 11.41     | 9.53     | 0.38       | 0.76      | 0.72       | 0.63        | 0.53      | 0.77     | 0.35       | 0.59         |
> | **MoE-SVD (512) +LoRA** | **4.29**        | **10.99** | **8.81** | **0.39**   | **0.81**  | **0.74**   | **0.65**    | **0.54**  | **0.79** | **0.36**   | **0.61**     |
>
>
>
> **(2)** Model compression methods (including ours and compression methods) make tradeoffs of performance-efficiency and always introduces the performance degradation to reduce model size, **which is common phenomenon in this field**. Our benefits are that our MoE-SVD effectively reduces performance drops than other methods: noticeable performance-efficiency improvements compared to other structured compression methods (see Table below),  significant performance gains (>100 lower perplexity and 5%~20% accuracy gains) than to other SVD methods (see Table 2).

---

> ### Author Response · Authors · 2024-11-22
> **Response to Reviewer nYc5  (Part 2/5)**
>
> **Table:  More comparisons with other structured compression methods under around 20% compression ratio.**
>
> | Mixtral-8×7B (20%)   | speed up | Token/Sec | WikiText-2↓ | PTB↓      | C4↓      | Openb.   | ARC_e    | WinoG.   | HellaS.  | ARC_c    | PIQA     | MathQA   | Average↑ |
> | -------------------- | -------- | --------- | ----------- | --------- | -------- | -------- | -------- | -------- | -------- | -------- | -------- | -------- | -------- |
> | Wanda (2:4)          | 1.04x    | 91.1      | 4.72        | 18.8      | 8.43     | 0.32     | 0.76     | 0.72     | 0.55     | 0.47     | 0.79     | 0.36     | 0.57     |
> | SparseGPT (2:4)      | 1.06x    | 93.1      | 4.61        | 21.11     | 8.19     | 0.3      | 0.77     | 0.74     | 0.56     | 0.45     | 0.77     | 0.35     | 0.56     |
> | LoSparse             | 1.06x    | 92.9      | 953.51      | 805.16    | 1273.12  | 0.2      | 0.27     | 0.49     | 0.28     | 0.26     | 0.53     | 0.2      | 0.32     |
> | MC-SMoE [1]          | 1.09x    | 95.3      | 1341.36     | 1316.52   | 1478.13  | 0.26     | 0.28     | 0.51     | 0.29     | 0.25     | 0.54     | 0.19     | 0.33     |
> | Unified-MoE-Compress | 1.13x    | 99.1      | 6.12        | 14.67     | 11.61    | 0.30     | 0.73     | 0.70     | 0.54     | 0.46     | 0.73     | 0.33     | 0.54     |
> | **MoE-SVD**          | **1.2x** | **104.7** | **4.44**    | **15.21** | **8.32** | **0.32** | **0.78** | **0.73** | **0.57** | **0.48** | **0.79** | **0.37** |          |
>
>
>
> **(3)** In addition to accuracy preservation, our superiority in **inference acceleration also deserves to be considered for efficiency performance across many mobile or speed-demanding scenarios. Moreover, our MoE-SVD are hardware-independent and successfully compact LLM to high compression ratios without additional training.**
>
> ------
>
> **Q2: Comparison with fine-tuning compression methods like  MC-SMoE [1].**
>
> **A2**:  (1) As shown in Table below, our approach differs from MC-SMoE [1] in comparison methods, algorithmic pipeline,  decomposition  and training approach,  evaluating Models:  Positioned as a model merge technique, MC-SMoE [1]  only addresses small-scale MoE by first merging experts and then applying vanilla decomposition before fine-tuning.  In contrast,  our MoE-SVD designs customized strategies like selective decomposition for large-scale MoEs without expert merging and fine-tuning.
>
>
>
> **Table:  Comparisons with MC-SMoE [1].**
>
> | Method      | Comparison Methods  | Pipeline             | Decomposition          | Training          | Advanced strategy for MoE LLMs                               | Evaluating Models                                            |
> | ----------- | ------------------- | -------------------- | ---------------------- | ----------------- | ------------------------------------------------------------ | ------------------------------------------------------------ |
> | MC-SMoE [1] | model merge methods | merge, then Compress | low-rank+sparse matrix | needs fine-tuning | frequency  merge                                             | t5-base (220M),switch-base-32 (2.0B)                         |
> | MoE-SVD     | model SVD methods   | decompose, then trim | low-rank matrix        | with fine-tuning  | activation-weighted SVD, selective decomposition, matrix sharing-trimming | Mixtral-8×7B\|22B, Phi-3.5-MoE(41.9B), and DeepSeekMoE (16.4B) |
>
>  (1) As shown in Table below,  our method MoE-SVD demonstrates superior speed improvements compared to MC-SMoE, achieving a 1.2x speedup and significantly higher performance across various benchmark datasets, including WikiText-2, PTB, and C4. These performance gains highlight the enhanced efficiency and scalability of MoE-SVD in processing large-scale MoE tasks, contributing to improved model performance and throughput.
>
>
>
> **Table:  More comparisons with MC-SMoE [1] under around 20% compression ratio.**
>
> | Mixtral-8×7B (20%) | Speed up | Token/Sec | WikiText-2↓ | PTB↓      | C4↓      | Openb.   | ARC_e    | WinoG.   | HellaS.  | ARC_c    | PIQA     | MathQA   | Average↑ |
> | ------------------ | -------- | --------- | ----------- | --------- | -------- | -------- | -------- | -------- | -------- | -------- | -------- | -------- | -------- |
> | MC-SMoE [1]        | 1.09x    | 95.3      | 1341.36     | 1316.52   | 1478.13  | 0.26     | 0.28     | 0.51     | 0.29     | 0.25     | 0.54     | 0.19     | 0.33     |
> | **MoE-SVD**        | **1.2x** | **104.7** | **4.44**    | **15.21** | **8.32** | **0.32** | **0.78** | **0.73** | **0.57** | **0.48** | **0.79** | **0.37** | **0.58** |
>
>
> ----

---

> > ### Author Response · Authors · 2024-11-22
> > **Response to Reviewer nYc5 (Part 3/4)**
> >
> > **Q3: About using AFM instead of SVD for decomposition.**
> >
> > **A3:** (1) As stated in Section 3.1 [Line 212-215], we would like to clarify that our approach employs an activation-weighted SVD (see Algorithm below), rather than the vanilla SVD. With activation matrix $X$ and original weight $W_original$, we compute the activation-weighted matrix by scaling the original weight matrix based on the activation statistics: $Waw = W_original ·S$,  where $W_aw$  represents the activation-weighted matrix and matrix $S$ is obtained through
> > cholesky decomposition of activation gram matrix $X\cdot X^T$. We then perform SVD on Waw and finalcompressed weight matrix is obtained by truncating the smallest singular values:
> > $$
> > W_{\text{aw}}  = U\cdot \text{Trunc}(\Sigma) \cdot V^T \cdot S^{-1},
> > $$
> >
> >
> > This activation-weighted approach effectively mitigates reconstruction loss from outliers during matrix decomposition while maintaining the essential characteristics of the original weight distribution. Note that we explicitly acknowledge the strategy's origin from previous works such as SVD-LLM and ASVD [Line 212, 988], emphasizing that we do not present it as our original contribution. This setting holds particular significance for large-scale LLM applications, as without it, the outcomes of LLM decomposition tend to exhibit considerable weaknesses, often exceeding 100 perplexity points. **We have added more details in Appendix D.2  [Line 941-965] and pyhon pseudo-code Algorithm 1  [Line 1080-1118] in  the revision .**
> >
> > **Algorithm: Python pseudo-code code for SVD Expert Decomposition of MoE-SVD.**
> >
> >
> > ```python
> > class MoESVDCompression:
> >     def __init__(self, truncate_k=None, top_k_experts=2):
> >         """
> >         Initialize Activation-Weighted SVD with Matrix Sharing and Trimming
> >
> >         Args:
> >             truncate_k (int): Number of singular values to keep
> >             top_k_experts (int): Number of top experts to select for U-matrix trimming
> >         """
> >         self.truncate_k = truncate_k
> >         self.top_k_experts = top_k_experts
> >
> >     def compute_activation_weights(self, X):
> >         """
> >         Compute activation-weighted scaling matrix S using Cholesky decomposition
> >
> >         Args:
> >             X (torch.Tensor): Activation matrix [batch_size,feature_dim]
> >             torch.mm(X, X.t()) is Cumulative activation matrix, representing the sum of processed activation data.
> >         """
> >         # Compute Gram matrix
> >         gram = torch.mm(X, X.t())
> >
> >         # Cholesky decomposition
> >         S = torch.linalg.cholesky(gram)
> >         return S
> >
> >     def decompose_expert(self, W_original, X):
> >         """
> >         Perform activation-weighted SVD on single expert
> >
> >         Args:
> >             W_original (torch.Tensor): Original weight matrix
> >             X (torch.Tensor): Activation matrix
> >         """
> >         # Compute activation-weighted matrix
> >         S = self.compute_activation_weights(X)
> >         W_aw = torch.mm(W_original, S)
> >
> >         # Perform SVD
> >         U, sigma, V = torch.linalg.svd(W_aw, full_matrices=False)
> >
> >         # Truncate if specified
> >         if self.truncate_k is not None:
> >             U = U[:, :self.truncate_k]
> >             sigma = sigma[:self.truncate_k]
> >             V = V[:self.truncate_k, :]
> >
> >         return U, torch.diag(sigma), V, S
> >
> > ```
> >
> >
> >
> > **(2)** While AFM is an effective method that performs PCA on the covariance matrix to reconstruct an approximate matrix, our activation-weighted SVD approach also considers the impact of outliers during matrix decomposition. Both techniques ultimately aim to address the influence of outliers in the matrix decomposition process. **We believe that AFM can achieve similar performance** and have considered experimenting with it. However, we experience difficulties because AFM lacks an open-source code and there are limits on rebuttal time. We will explore this in future work.
> >
> > -----
> >
> > **Q4:  About speed performance of MOE-SVD on DeepSeekMoE.**
> >
> > **A4** : **(1)** Following the suggestion, we performed evaluation of MoE-SVD on DeepSeekMoE with  shared experts  in Table below. Our experiments show tha**t MoE-SVD effectively reduces the model size and computational requirements for DeepSeekMoE, leading to inference speedups.** Specifically, MoE-SVD also compresses the shared expert components, alleviating the inference bottleneck without substantial loss in performance.
> >
> > **Table: Metrics (model size, TFLOPs, runtime) of MoE-SVD on DeepSeekMoE. Runtime denotes runtime throughput (Tokens/sec)**
> >
> > | DeepSeekMoE | Dense    | 20%      | 30%      | 40%      | 50%      | 60%      |
> > | ----------- | -------- | -------- | -------- | -------- | -------- | -------- |
> > | model size  | 16.4B    | 13.2B    | 11.4B    | 9.7B     | 8B       | 6.4B     |
> > | TFLOPs      | 1.10e+14 | 1.02e+14 | 9.88e+13 | 9.29e+13 | 9.25e+13 | 8.82e+13 |
> > | Runtime     | 52.53    | 62.79    | 94.71    | 118.93   | 119.81   | 128.71   |

---

> ### Author Response · Authors · 2024-11-22
> **Response to Reviewer nYc5 (Part 4/5)**
>
> **(2)** Following the suggestion, we involve extra experiment on Qwen2-57B-A14B as in following Table.   The results demonstrate consistent performance and speed improvements across these diverse tasks, suggesting our method's generalizability.  Thanks for the suggestion, we already added this experiment in the revision **[Line 857-863]**.
>
> **Table:  Performance of Qwen2-57B-A14B compressed by MoE-SVD under 20% compression ratios.**
>
> | Models                                | Runtime          | WikiText-2↓ | PTB↓      | C4↓       | Openb.   | ARC_e    | WinoG.   | HellaS.  | ARC_c    | PIQA     | MathQA   | Average↑ |
> | ------------------------------------- | ---------------- | ----------- | --------- | --------- | -------- | -------- | -------- | -------- | -------- | -------- | -------- | -------- |
> | Qwen2-57B-A14B  Original              | 42.70            | 4.32        | 11.66     | 9.23      | 0.33     | 0.75     | 0.74     | 0.63     | 0.47     | 0.8      | 0.39     | 0.58     |
> | **Qwen2-57B-A14B MoE-SVD (20%)**      | **53.16(x1.25)** | **6.52**    | **14.61** | **13.64** | **0.29** | **0.71** | **0.69** | **0.58** | **0.42** | **0.74** | **0.33** | **0.53** |
> | **Qwen2-57B-A14B MoE-SVD (20%)+LoRA** | **53.16(x1.25)** | **5.41**    | **13.26** | **11.63** | **0.30** | **0.74** | **0.73** | **0.61** | **0.45** | **0.78** | **0.35** | **0.56** |
>
>
>
> **Q5: Tensor parallelism and potential communication overhead.**
>
> **A5**: **(1)** Our MoE-SVD can reduce the computation time because of the reduced size of the weight matrix  via decomposition. As shown in Table below, our method can get 1.2x~1.5x speedup in inference. These speedup results already outperform other structured compression at algorithmic level, which confirms the effectiveness of our method.
>
> **Table: Metrics (model size, TFLOPs, runtime) of MoE-SVD. Runtime denotes runtime throughput (Tokens/sec).**
>
> | Mixtral-8×7B   | Dense     | 20%      | 30%      | 40%      | 50%      | 60%      |
> | -------------- | --------- | -------- | -------- | -------- | -------- | -------- |
> | model size     | 46.7B     | 37.1B    | 32.2B    | 28.3B    | 23B      | 17.6     |
> | TFLOPS         | 5.27e+14  | 4.92e+14 | 4.40e+14 | 4.26e+14 | 3.97e+14 | 3.67e+14 |
> | runtime        | 87.73     | 104.66   | 106.03   | 108.83   | 123.88   | 156.10   |
> | **Phi3.5 moe** | **Dense** | **20%**  | **30%**  | **40%**  | **50%**  | **60%**  |
> | model size     | 41.9B     | 33.2B    | 29B      | 25.2B    | 20.6B    | 16.4B    |
> | TFLOPS         | 2.72e+14  | 2.46e+14 | 2.27e+14 | 2.00e+14 | 1.82e+14 | 1.71e+14 |
> | runtime        | 98.2      | 108.63   | 114.8    | 124.79   | 137.17   | 148.7    |
>
>
>
> **(2)** Combining with particular hardware system optimization methods like tensor parallelism, all the low-rank decomposition approaches (including our methods. MC-SMoE [1], ASVD and SVD-LLM, etc) will introduce communication overheads. This is **a common phenomenon for this kind of techniques** in specific scenarios like tensor parallelism. But we highlight that the t**otal overhead optimization represents the trade-off between computation time and communication time**, which also depends on the individual hardware device.

---

> ### Author Response · Authors · 2024-11-22
> **Response to Reviewer nYc5 (Part 5/5)**
>
> **(3) Analysis of trade-off between computation time and communication time:**
>
> Our SVD-MoE methodology introduces a sophisticated approach to matrix decomposition in expert systems, fundamentally transforming the computational architecture while preserving mathematical integrity. The core operation involves decomposing the expert weight matrix $\mathbf{W} \in \mathbb{R}^{H \times H}$ (with hidden dimension $H = 4096$) into constituent matrices $\mathbf{U} \in \mathbb{R}^{H \times r}$ and $\mathbf{V}^\top \in \mathbb{R}^{r \times H}$, where the reduced rank $r = 1024$ represents a quarter of the hidden dimension ($r = H/4$). For input activations $\mathbf{X} \in \mathbb{R}^{B \times H}$ with batch size $B = 128$, the computational pathway evolves from the direct multiplication $\mathbf{Y = XW}$ to a two-stage process $\mathbf{Y = XUV}^\top$, introducing an intermediate activation $\mathbf{Z = XU} \in \mathbb{R}^{B \times r}$.
>
>
>
> **The computational efficiency** manifests in quantifiable metrics:
>
> $\text{Computations}_{\text{before}} = B \times H^2 = 128 \times 4096^2 = 2.15 \times 10^{11}$
>
> $\text{Computations}_{\text{after}} = 2 \times B \times H \times r = 2 \times 128 \times 4096 \times 1024 = 1.07 \times 10^{11}$
>
> Which achieving a 50% reduction.
>
> This efficiency gain, however, introduces additional complexity in tensor parallelism operations. The communication volume expands from $B \times H = 128 \times 4096 = 524,288$ elements to $B \times (r + H) = 128 \times (1024 + 4096) = 655,360$ elements, representing a 25% increase.
>
>
>
> **Performance metrics** on A100 GPUs (600 GB/s NVLink bandwidth, FP32 precision):
>
> Communication time, calculated as $\text{Communication Time} = \frac{\text{Total Data (bytes)}}{\text{Bandwidth (bytes/s)}}$, increases from $3.5 \times 10^{-6}$ seconds to $4.37 \times 10^{-6}$ seconds.
>
>
>
> **Computation-Communication Tradeoff**:
>
> Computation time, based on 19.5 TFLOPS peak performance, decreases substantially from $\frac{2.15 \times 10^{11}}{19.5 \times 10^{12}} \approx 11 \times 10^{-3}$ seconds to $\frac{1.07 \times 10^{11}}{19.5 \times 10^{12}} \approx 5.5 \times 10^{-3}$ seconds. The aggregate processing time improves from $11.0035 \times 10^{-3}$ seconds to $5.50437 \times 10^{-3}$ seconds, yielding a net optimization of $5.49913 \times 10^{-3}$ seconds.
>
> **The marginal increase in communication overhead ($\Delta \text{Communication Time} = 0.87 \times 10^{-6}$ seconds) proves negligible against the substantial computational savings ($\Delta \text{Computation Time} = 5.5 \times 10^{-3}$ seconds).**
>
> This favorable ratio demonstrates that the SVD-MoE architecture successfully navigates the trade-off between computational efficiency and communication overhead. The high-bandwidth interconnect infrastructure effectively mitigates the increased communication volume, ensuring that the additional synchronization requirements do not compromise the overall performance gains achieved through matrix decomposition.
>
> **(4)**  We had considered adding experiments. However, we were unable to perform experiments within the rebuttal period. PyTorch's tensor parallelism currently supports only official models. Adapting it to non-official models like ours would require modifications to the underlying library, which was not feasible in the given timeframe. Nonetheless, our analysis strongly indicates that the benefits of reduced computation outweigh the additional communication costs. We will explore this in future work.
>
> ------
>
> **Finally, we genuinely hope that our explanations and efforts can improve the overall evaluation of our work.** We are glad to discuss further comments and suggestions.

---

### Official Review · Reviewer_eFn1 · 2024-10-31

**Soundness:** 3
**Presentation:** 3
**Contribution:** 3
**Rating:** 6
**Confidence:** 4

**Summary:**

This paper proposed MoE-SVD, which is a training-free decomposition-based compression framework. Specifically, they first decompose experts into low-rank matrices via SVD. In particular, they selectively decompose the expert layers based on sensitivity metrics. Then, they share a single V-matrix across all experts and use a top-k selection for U-matrices for parameter reduction.

**Strengths:**

1. This paper explores SVD-based methods for MoE LLMs and identifies key factors that degrade performance: certain layers are sensitive to decomposition, and activation patterns differ in MoE, revealing expert redundancy.
2. the authors propose a selective decomposition strategy along with low-rank matrix sharing and trimming, which are well-justified approaches.
3. Overall, the paper is clear and easy to follow.

**Weaknesses:**

1. In line 53, could you elaborate on "Hardware Dependency" in the context of semi-structured sparse methods?

2. In Eq. (4),  $r_i$ represents the rank, but I believe the matrix should be of full rank, so the purpose of using $r_i$ here is unclear. Additionally, what does $f_i$ represent? Is it related to sampling frequency?

3. Figure 4 is unclear to interpret. It appears that the green grid denotes the decomposed experts, but could you clarify what the compression ratios, also shown on the y-axis, represent? While the overall motivation, that the approach seeks to identify layers with varying decomposition sensitivity, is understandable, the details require further clarification.

4. For the experiments, including Qwen in the experiments could strengthen the results.

**Questions:**

Please clarify more on Section 3.2

---

> ### Author Response · Authors · 2024-11-22
> **Response to Reviewer eFn1 (Part1/2)**
>
> **Dear Reviewer eFn1,**
>
> Thanks for constructive comments. We have tried our best to address all concerns in the last few days. If the reviewer finds our response adequate, **we would appreciate it if the reviewer considers raising the score**. Please see our responses below one by one：
>
> ------
>
> **Q1: About "Hardware Dependency" for semi-structured sparse methods.**
>
>  **A1:** Up to now, only NVIDIA Ampere and NVIDIA Hopper architecture GPUs  include new sparse Tensor Cores that can support semi-structured sparse methods [1]. Commonly used NVIDIA Volta (e.g., V100), Turing (e.g., RTX 2080 Ti), Pascal, Maxwell, and Kepler series architecture GPUs do not support semi-structured sparse methods.   Thanks for the suggestion, we have added this in the revision [Line 53].
>
> [1] NVIDIA Corporation. (2020). NVIDIA Ampere Architecture In-Depth. *NVIDIA Developer Blog*.
>
> -------
>
> **Q2 & Q5: About variables in Equation 4 and  more clarify  on Section 3.2.**
>
> **A2:** **(1)  The variable $r_i$ denotes the principal rank of the $i$-th expert, which is the number of dominant singular values in the diagonal matrix $\Sigma_i$ obtained from the SVD of the expert's weight matrix $W_i$.**   The  **principal rank  $r_i$ is our self-defined as the number of singular values in $\Sigma_i$ that exceed a given threshold**, effectively capturing the dimensionality of the weight matrix's significant components. This rank reflects the structural complexity of the expert's weight representation, with higher values of $r_i$ indicating more complex and information-rich weights.  **Note that  $r_i$ does not denote vanilla rank of the matrix (number of non-zero singular values)**.
>
> **(2)** **The $f_i$ represents the sampling frequency of the $i$-th expert**, quantifying the frequency with which this expert is selected by the router during inference. This metric also reflects the number of selections made by the expert within the calibration dataset. It is computed based on the routing decisions of the gating network $G(x)$ across a representative dataset $\mathcal{X}$ :
> $$
> f_i = \frac{\sum_{x \in \mathcal{X}} \mathbb{I}[i \in \text{TopK}(G(x), k)]}{|\mathcal{X}|},
> $$
>
> where $\mathbb{I}[\cdot]$ is the indicator function, and $k$ is the number of experts selected by the top-$k$ gating mechanism.
>
>
>
> **(3) The  $a_i$ signifies the activation outliers for the $i$-th expert**, defined as the outlier ratio within the expert's weight matrix. Outliers are identified as values exceeding a predefined threshold, which is set as a multiple of the mean value of the matrix. Mathematically, the outlier ratio is calculated as:
>
> $$
> a_i = \frac{\sum_{a \in A_i} \mathbb{I}(|a| > \tau \cdot \operatorname{Mean}(|A_i|))}{|A_i|}
> $$
>
>
> where $|A_i|$ denotes the total number of activations for the $i$-th expert, $\operatorname{Mean}(|A_i|)$ is the mean absolute value of these activations, and $\tau$ is a user-defined threshold. This metric highlights the presence of outlier activations indicative of the expert's contribution to the model's capacity.
>
> **(4)  Ablation studies of  sensitivity metric.** Our metric of $r_i$ and $a_i$  weighted by $f_i$ provides a comprehensive metric for identifying which expert layers are most suitable for decomposition.  Results in  Table 3 show that our metrics can obtain better performance (8.67 vs.16.57 ) than vanilla $a_i$ (Non-uniform SVD [OWL]) . Our additional experiments  show that our metrics with $r_i$ and $f_i$ obtained 9.85 and 12.65, respectively, demonstrating the importance of these designs.
>
> **Thanks for the suggestion, we have clarified Section 3.2 [Line 256-233], added more details in Appendix D.2  [Line 941-965] and pyhon pseudo-code Algorithm 1  [Line 1080-1118] in  the revision .**

---

> > ### Author Response · Authors · 2024-11-22
> > **Algorithm: Pyhon pseudo-code for sensitivity metric of MoE-SVD**
> >
> > ```pseudocode
> > def compute_layer_sensitivity(experts_weights, activations, gating_outputs, calibration_data, top_k=2, tau=2.0):
> >     """
> >     Compute layer-wise sensitivity metric S_L for MoE compression
> >
> >     Args:
> >         experts_weights (list of torch.Tensor): Weight matrices for each expert
> >         activations (list of torch.Tensor): Activation values for each expert
> >         gating_outputs (torch.Tensor): Router outputs for calibration data
> >         calibration_data (torch.Tensor): Calibration dataset
> >         top_k (int): Number of experts to select per token
> >         tau (float): Threshold for activation outliers
> >
> >     Returns:
> >         float: Layer sensitivity score S_L
> >     """
> >     num_experts = len(experts_weights)
> >     device = experts_weights[0].device
> >
> >     # Compute sampling frequency (f_i)
> >     top_k_indices = torch.topk(gating_outputs, top_k, dim=-1).indices
> >     expert_counts = torch.zeros(num_experts, device=device)
> >     for indices in top_k_indices:
> >         expert_counts[indices] += 1
> >     f_i = expert_counts / len(calibration_data)
> >
> >     # Compute principal rank (r_i) using SVD
> >     r_i = torch.zeros(num_experts, device=device)
> >     for i, weight in enumerate(experts_weights):
> >         U, S, V = torch.linalg.svd(weight)
> >         # Count singular values above threshold
> >         threshold = torch.max(S) * 1e-2  # Threshold
> >         r_i[i] = torch.sum(S > threshold)
> >
> >     # Compute activation outliers (a_i)
> >     a_i = torch.zeros(num_experts, device=device)
> >     for i, activation in enumerate(activations):
> >         mean_abs_act = torch.mean(torch.abs(activation))
> >         outliers = torch.sum(torch.abs(activation) > tau * mean_abs_act)
> >         a_i[i] = outliers / activation.numel()
> >
> >     # Compute final sensitivity metric S_L
> >     S_L = torch.sum(f_i * r_i * a_i)
> >
> >     return S_L
> >
> > ```

---

> > > ### Author Response · Authors · 2024-11-22
> > > **Response to Reviewer eFn1 (Part 2/2)**
> > >
> > > --------
> > >
> > > **Q3: About interpretation on Figure 4.**
> > >
> > > **A3:** Following the suggestion, we adopt following Table including detailed numbers alternative to Figure for clearer illustration of decomposition results. These results indicate clear trends: as the compression ratio increases, more layers are decomposed in both models. Specifically, middle layers are more likely to be decomposed, while the first and last layers tend to be retained. These decomposition patterns align well with empirical observations on MoE LLMs and our sensitivity decomposition scores derived from the experiments. Thanks for the suggestion, we have clarified this in the revision [Line 234-252].
> > >
> > > **Table :  Decomposed layers of our selective decomposition strategy by varying compression ratios .**
> > >
> > > | Mixtral-8×7B       |                                                            |
> > > | ------------------ | ---------------------------------------------------------- |
> > > | Compression ratios | Selected Decomposition Layers                              |
> > > | 20%                | [3,5,6,7,9,12,23,24,25],                                   |
> > > | 30%                | [3,5,6,7,9,12,13,22,23,24,25,26],                          |
> > > | 40%                | [3,5,6,7,9,10,12,13,21,22,23,24,25,26],                    |
> > > | 50%                | [3,5,6,7,9,10,12,13,14,15,16,20,21,22,23,24,25,26],        |
> > > | 60%                | [2,3,5,6,7,9,10,12,13,14,15,16,17,20,21,22,23,24,25,26,27] |
> > > | **Phi3.5 MoE**     |                                                            |
> > > | Compression ratios | Selected Decomposition Layers                              |
> > > | 20%                | [11,12,15,20,21,23,24,25],                                 |
> > > | 30%                | [11,12,15,20,21,23,24,25,27,28],                           |
> > > | 40%                | [10,11,12,15,16,18,20,21,23,24,25,26,27,28],               |
> > > | 50%                | [5,6,10,11,12,13,15,16,18,20,21,23,24,25,26,27,28],        |
> > > | 60%                | [5,6,10,11,12,13,15,16,18,19,20,21,23,24,25,26,27,28,29]   |
> > >
> > > ------
> > >
> > > **Q4: About  experiments on Qwen model.**
> > >
> > > **A4:** (1) Following the suggestion, we involve extra experiment on Qwen2-57B-A14B as in following Table.   The results demonstrate consistent performance improvements across these diverse tasks, suggesting our method's generalizability.  Thanks for the suggestion, we already added this experiment in the revision **[Line 857-863]**.
> > >
> > > (2) Our current experimental setup already includes a comprehensive set of modern MoE models including **Mixtral-8×7B|22B, Phi-3.5-MoE, and DeepSeekMoE**, providing strong validation across different architectures.
> > >
> > > **Table :  Performance of Qwen2-57B-A14B compressed by MoE-SVD under 20% compression ratios.**
> > >
> > > | Models                                | Runtime Token/Sec | WikiText-2↓ | PTB↓      | C4↓       | Openb.   | ARC_e    | WinoG.   | HellaS.  | ARC_c    | PIQA     | MathQA   | Average↑ |
> > > | ------------------------------------- | ----------------- | ----------- | --------- | --------- | -------- | -------- | -------- | -------- | -------- | -------- | -------- | -------- |
> > > | Qwen2-57B-A14B  Original              | 42.70             | 4.32        | 11.66     | 9.23      | 0.33     | 0.75     | 0.74     | 0.63     | 0.47     | 0.8      | 0.39     | 0.58     |
> > > | **Qwen2-57B-A14B MoE-SVD (20%)**      | **53.16(x1.25)**  | **6.52**    | **14.61** | **13.64** | **0.29** | **0.71** | **0.69** | **0.58** | **0.42** | **0.74** | **0.33** | **0.53** |
> > > | **Qwen2-57B-A14B MoE-SVD (20%)+LoRA** | **53.16(x1.25)**  | **5.41**    | **13.26** | **11.63** | **0.30** | **0.74** | **0.73** | **0.61** | **0.45** | **0.78** | **0.35** | **0.56** |
> > >
> > > ----
> > >
> > > **Finally,** we hope our response could address the concerns, and we thank the reviewer again for the helpful comments. We are glad to discuss further comments and suggestions.

---

### Official Review · Reviewer_ghaz · 2024-11-02

**Soundness:** 2
**Presentation:** 2
**Contribution:** 2
**Rating:** 3
**Confidence:** 4

**Summary:**

This paper introduces MoE-SVD, a novel framework leveraging SVD for compressing Mixture-of-Experts architectures in large language models. MoE-SVD addresses key challenges of decomposition collapse and parameter redundancy in MoE models, utilizing techniques such as selective decomposition, matrix sharing, and matrix trimming. Experiments conducted on Mixtral-8x7B, Phi-3.5-MoE, and DeepSeekMoE demonstrate its ability and inference acceleration. Further evaluations with LoRA fine-tuning and quantization suggest MoE-SVD’s adaptability across different MoE backbones.

**Strengths:**

- The paper thoroughly evaluates the proposed method across various MoE architectures, including Mixtral-8x7B, Phi-3.5-MoE, and DeepSeekMoE.
- Additional experiments with LoRA fine-tuning and quantization validate the efficacy of MoE-SVD.
- MoE-SVD introduces a unique method for MoE compression. The matrix-sharing techniques can be interesting.

**Weaknesses:**

- The paper does not include comparisons with structured compression methods directly applicable to MoE models, such as those presented in [1, 2, 3]. Additionally, it overlooks methods that specifically target expert layer compression or simultaneously address both expert layer and expert number compression, as found in [4, 5, 6]. These comparisons are crucial for a comprehensive evaluation of MoE-SVD's performance.
- Key metrics such as model size reduction, TFLOPS, and runtime are missing. These metrics are critical for assessing the practical efficiency of the proposed method.
-  MoE-SVD shows a notable decline in performance even with a 20% parameter reduction, indicating possible limitations in maintaining model quality during compression. Furthermore, the absence of significance testing for performance claims weakens the robustness of the results.
- The representation and decomposition process of the expert matrix is ambiguous. In the expert decomposition subsection, the paper describes the expert matrix as $\mathbb{R}^{m \times n}$, but experts are typically two- or three-layer neural networks. It is unclear whether the matrix is decomposed individually for each layer or concatenated before decomposition, and if so, how the concatenation is done. Clarifying these points is important for the reader's understanding.
- The calculation of the sensitivity score in the selective decomposition strategy lacks clarity. It is unclear how activations are computed, how the dataset for expert frequency calculation is chosen, and whether the frequency values obtained from one dataset can be generalized to others. Providing pseudo-code would enhance comprehensibility.
- The reasoning behind sharing the V-matrix to maintain performance is insufficiently explained. Further elaboration on the properties that enable the effective shared use of this matrix would be beneficial.
-  The rationale for each expert storing information from two other experts, as indicated in equation (8), requires clearer justification. Specifically, an explanation is needed on why each expert needs information from other experts and how this contributes to enhancing the model’s overall performance. Although the paper mentions diversity as a factor, it is unclear why simply combining the U-matrix suffices can work, especially in the context of zero-shot MoE-LLMs.

[1] A simple and effective pruning approach for large language models. ICLR 2024

[2] Sparsegpt: Massive language models can be accurately pruned in one-shot. ICML 2023

[3] LoSparse: Structured Compression of Large Language Models based on Low-Rank and Sparse Approximation. ICML 2023

[4] Merge, then compress: Demystify efficient SMoe with hints from its routing policy. ICLR 2024

[5] STUN: Structured-Then-Unstructured Pruning for Scalable MoE Pruning.

[6] Demystifying the Compression of Mixture-of-Experts Through a Unified Framework.

**Questions:**

Please refer to the weakness part.

---

> ### Author Response · Authors · 2024-11-22
> **Response to Reviewer ghaz (Part1/3)**
>
> **Dear Reviewer ghaz,**
>
> Thanks for constructive comments. We have tried our best to address all concerns in the last few days. If the reviewer finds our response adequate, **we would appreciate it if the reviewer considers raising the score**. Please see our responses below one by one：
>
> ------
>
> **Q1: About  comparisons with structured compression methods [1]~[6].**
>
> **A1: (1)** Following the suggestions, we have conducted additional experiments to compare our MoE-SVD method with these structured compression methods in Table below. Note that we were unable to include STUN [5] in our comparisons, as it was recently published in September on arXiv without available code.  **We already involve this  comparisons [Line 833-845] in the revision.**
>
> **Table:  More comparisons with other structured compression methods under around 20% compression ratio.**
>
> | Mixtral-8×7B (20%)   | Speed up↑ | Token/Sec↑ | WikiText-2↓ | PTB↓      | C4↓      | Openb.   | ARC_e    | WinoG.   | HellaS.  | ARC_c    | PIQA     | MathQA   | Average↑ |
> | -------------------- | -------- | --------- | ----------- | --------- | -------- | -------- | -------- | -------- | -------- | -------- | -------- | -------- | -------- |
> | Wanda (2:4)          | 1.04x    | 91.1      | 4.72        | 18.8      | 8.43     | 0.32     | 0.76     | 0.72     | 0.55     | 0.47     | 0.79     | 0.36     | 0.57     |
> | SparseGPT (2:4)      | 1.06x    | 93.1      | 4.61        | 21.11     | 8.19     | 0.3      | 0.77     | 0.74     | 0.56     | 0.45     | 0.77     | 0.35     | 0.56     |
> | LoSparse             | 1.06x    | 92.9      | 953.51      | 805.16    | 1273.12  | 0.2      | 0.27     | 0.49     | 0.28     | 0.26     | 0.53     | 0.2      | 0.32     |
> | Merge, then compress | 1.09x    | 95.3      | 1341.36     | 1316.52   | 1478.13  | 0.26     | 0.28     | 0.51     | 0.29     | 0.25     | 0.54     | 0.19     | 0.33     |
> | Unified-MoE-Compress | 1.13x    | 99.1      | 6.12        | 14.67     | 11.61    | 0.30     | 0.73     | 0.70     | 0.54     | 0.46     | 0.73     | 0.33     | 0.54     |
> | **MoE-SVD**          | **1.2x** | **104.7** | **4.44**    | **15.21** | **8.32** | **0.32** | **0.78** | **0.73** | **0.57** | **0.48** | **0.79** | **0.37** | **0.58** |
>
>
>
> **(2)** The result shows that our MoE-SVD has significant performance gains over LoSparse [3] and MC-SMoE [4] in both speedup and performance metrics. This improvement stems from our customized strategies for MoE compression, such as selective decomposition, which extend beyond the straightforward low-rank decomposition utilized by LoSparse and MC-SMoE.
>
> **(3)** Compared to prunning methods in structured settings in Wanda [1] and SparseGPT [2], our approach achieves significant speedups and consistent performance gains.
>
> **(4)** We have discussed the drawbacks of these pruning methods, like Wanda and and SparseGPT, in original manuscript [Line 46-69], noting their dependence on specific hardware and their limited speedup ratios.  Our MoE-SVD designs customized strategies like selective decomposition for large-scale MoEs without expert merging and fine-tuning, in contrast to MC-SMoE [4] , which only addresses small-scale MoE by first merging experts and then applying vanilla decomposition before fine-tuning (see addition discussion in the revision [Line 156-158]).
>
>
>
> **Q2: About metrics (model size, TFLOPS, runtime).**
>
>  **A2:** **(1)** We clarify that we already include **some metrics in the original manuscript like runtime throughput (tokens/sec) in Figure 5 and Memory usage (GB) in Figure 6**.  We also present analysis about these metrics in the Experiment Section [Line 430-466]. For example, our method obtains 1.53 ×  acceleration in  runtime throughput.
>
> **(2)** Following the suggestions to provide a comprehensive assessment, we organize key metrics in the following Table, **which has been added to the revision [Line 900-915]**. These results specifically details runtime  metrics across various compression ratios, with explicit measurement of **speedups ranging from 1.2× to 1.5×**.
>
> **Table: Metrics (model size, TFLOPs, runtime) of MoE-SVD. Runtime denotes runtime throughput (Tokens/sec) on a single H800 GPU.**
>
> | Mixtral-8×7B   | Dense     | 20%      | 30%      | 40%      | 50%      | 60%      |
> | -------------- | --------- | -------- | -------- | -------- | -------- | -------- |
> | Model-size     | 46.7B     | 37.1B    | 32.2B    | 28.3B    | 23B      | 17.6     |
> | TFLOPs         | 5.27e+14  | 4.92e+14 | 4.40e+14 | 4.26e+14 | 3.97e+14 | 3.67e+14 |
> | Runtime        | 87.73     | 104.66   | 106.03   | 108.83   | 123.88   | 156.10   |
> | **Phi3.5 MoE** | **Dense** | **20%**  | **30%**  | **40%**  | **50%**  | **60%**  |
> | Model-size     | 41.9B     | 33.2B    | 29B      | 25.2B    | 20.6B    | 16.4B    |
> | TFLOPs         | 2.72e+14  | 2.46e+14 | 2.27e+14 | 2.00e+14 | 1.82e+14 | 1.71e+14 |
> | Runtime        | 98.2      | 108.63   | 114.8    | 124.79   | 137.17   | 148.7    |

---

> ### Author Response · Authors · 2024-11-22
> **Response to Reviewer ghaz (Part  2/3)**
>
> ------
>
> **Q4: About  decomposition process of the expert matrix.**
>
> **A4:**  Yes. our decomposition process applies SVD individually to each fully connected layer within each expert, rather than concatenating the layers. Thanks for the suggestion, we have clarified this **in Section 3.1 [line 196-202]** and provided more explanations and pyhon pseudo-code Algorithm 2  [Line 1134-1174] in the revision.
>
> -----
>
> **Q5: About  calculation of the sensitivity score.**
>
> **A5:**   **(1)The  $a_i$ signifies the activation outliers for the $i$-th expert**, defined as the outlier ratio within the expert's weight matrix. Outliers are identified as values exceeding a predefined threshold, which is set as a multiple of the mean value of the matrix. Mathematically, the outlier ratio is calculated as:
> $$
> a_i = \frac{\sum_{a \in A_i} \mathbb{I}(|a| > \tau \cdot \operatorname{Mean}(|A_i|))}{|A_i|}
> $$
>
> where $|A_i|$ denotes the total number of activations for the $i$-th expert, $\operatorname{Mean}(|A_i|)$ is the mean absolute value of these activations, and $\tau$ is a user-defined threshold. This metric highlights the presence of outlier activations indicative of the expert's contribution to the model's capacity.
>
> **(2)** **The $f_i$ represents the sampling frequency of the $i$-th expert**, quantifying the frequency with which this expert is selected by the router during inference. This metric also reflects the number of selections made by the expert within the calibration dataset. It is computed based on the routing decisions of the gating network $G(x)$ across a representative dataset $\mathcal{X}$ :
> $$
> f_i = \frac{\sum_{x \in \mathcal{X}} \mathbb{I}[i \in \text{TopK}(G(x), k)]}{|\mathcal{X}|},
> $$
>
> where $\mathbb{I}[\cdot]$ is the indicator function, and $k$ is the number of experts selected by the top-$k$ gating mechanism.
>
> **(3) Generalizability.**  Our extra tests show that frequency values obtained from WikiText-2 dataset indeed can work well with other datasets (PTB and C4) with only ~0.1 perplexity variations. In addition, Our experiments test results in multiple datasets in Table 2 and ablation  analysis in Table 5 also confirm the generalizability of our Moe-SVD.
>
> **(4)**  **Ablation studies of  sensitivity metric.** Our metric of $r_i$ and $a_i$  weighted by $f_i$ provides a comprehensive metric for identifying which expert layers are most suitable for decomposition.  Results in  Table 3 show that our metrics can obtain better performance (8.67 vs.16.57 ) than vanilla $a_i$ (Non-uniform SVD [OWL]) . Our additional experiments  show that our metrics with $r_i$ and $f_i$ obtained 9.85 and 12.65, respectively, demonstrating the importance of these designs.
>
> **Thanks for the suggestion, we have clarified Section 3.2 [Line 256-233], added more details in Appendix D.2  [Line 941-965] and pyhon pseudo-code Algorithm 1  [Line 1080-1118] in  the revision .**
>
> **Algorithm: Pyhon pseudo-code for sensitivity metric of MoE-SVD.**
>
> ```pseudocode
> def compute_layer_sensitivity(experts_weights, activations, gating_outputs, calibration_data, top_k=2, tau=2.0):
>     """
>     Compute layer-wise sensitivity metric S_L for MoE compression
>
>     Args:
>         experts_weights (list of torch.Tensor): Weight matrices for each expert
>         activations (list of torch.Tensor): Activation values for each expert
>         gating_outputs (torch.Tensor): Router outputs for calibration data
>         calibration_data (torch.Tensor): Calibration dataset
>         top_k (int): Number of experts to select per token
>         tau (float): Threshold for activation outliers
>
>     Returns:
>         float: Layer sensitivity score S_L
>     """
>     num_experts = len(experts_weights)
>     device = experts_weights[0].device
>
>     # Compute sampling frequency (f_i)
>     top_k_indices = torch.topk(gating_outputs, top_k, dim=-1).indices
>     expert_counts = torch.zeros(num_experts, device=device)
>     for indices in top_k_indices:
>         expert_counts[indices] += 1
>     f_i = expert_counts / len(calibration_data)
>
>     # Compute principal rank (r_i) using SVD
>     r_i = torch.zeros(num_experts, device=device)
>     for i, weight in enumerate(experts_weights):
>         U, S, V = torch.linalg.svd(weight)
>         # Count singular values above threshold
>         threshold = torch.max(S) * 1e-2  # Threshold
>         r_i[i] = torch.sum(S > threshold)
>
>     # Compute activation outliers (a_i)
>     a_i = torch.zeros(num_experts, device=device)
>     for i, activation in enumerate(activations):
>         mean_abs_act = torch.mean(torch.abs(activation))
>         outliers = torch.sum(torch.abs(activation) > tau * mean_abs_act)
>         a_i[i] = outliers / activation.numel()
>
>     # Compute final sensitivity metric S_L
>     S_L = torch.sum(f_i * r_i * a_i)
>
>     return S_L
>
> ```

---

> > ### Author Response · Authors · 2024-11-22
> > **Response to Reviewer ghaz (Part 3/3)**
> >
> > ----
> >
> > **Q6: About reason behind sharing the V-matrix.**
> >
> > **A6:** **(1)** In the following Table, our analysis revealed that the V-matrices obtained from the decomposition of different experts exhibit high similarity and redundancy. We quantified this using centered kernel alignment (CKA) distances. **These results show that these V matrices are in great similarity and redundancy and could be further parameter-shared and trimmed out of redundancy.** These high CKA values indicate that the V-matrices across experts are highly similar, suggesting functional redundancy in their representations.  **We have added this analysis in Figure 1, introduction [Line 62, 84] and Section 3.3 [Line 257] in the revision.**
> >
> > **Table: The centered kernel alignment (CKA) similarity before/after decomposition for decomposed layer [3,5,6,7,9,12,23,24,25], in Mixtral-8×7B on WikiText-2.**
> >
> > | Decomposed Layer/CKA distance      | 3     | 5     | 6     | 7     | 9     | 12    | 23    | 24    | 25    |
> > | ---------------------------------- | ----- | ----- | ----- | ----- | ----- | ----- | ----- | ----- | ----- |
> > | Original Expert Matrix             | 0.279 | 0.334 | 0.335 | 0.341 | 0.338 | 0.329 | 0.273 | 0.269 | 0.21  |
> > | Decomposed U-matrix of each Expert | 0.304 | 0.320 | 0.327 | 0.329 | 0.329 | 0.331 | 0.309 | 0.307 | 0.305 |
> > | Decomposed V-matrix of each Expert | 0.998 | 0.984 | 1     | 1     | 1     | 0.989 | 0.978 | 0.999 | 0.983 |
> >
> > **(2)** By sharing the V-matrix **exploits similarities between expert outputs to minimize redundancy**, computational costs and maintain performance, a strategy grounded in observed expert similarities during training. This strategy leverages **the inherent similarities** in the representations learned by the experts, as supported by the aforementioned studies. This approach reduces model parameters while preserving crucial output transformations.
> >
> > **(3)** As illustrated in Table 2 and Ablation 4.3, our sharing V-matrix reduces model parameter size and allows us to maintain more sensitive layers to improve performance.
> >
> > -------
> >
> > **Q7: About  expert information sharing.**
> >
> > **A7:**  **(1) As shown in Table of A6**, the similarity of decomposed U-matrices is similar to the original matrix, confirming that the decomposed U-matrices preserve diversity effectively. This is attributed to the fact that we multiply the singular vectors after the decomposition all into the U-matrix. According to the theory of SVD decomposition, singular vectors contain the properties of the original matrices, Thus, our U-matrix containing singular vectors maximally guarantees the unique information of the original matrices.
> >
> > **(2)** Considering that the decomposed V-matrices are highly similar after the decomposition, it is natural to remove the remaining ones for parameter efficiency. Because these V matrices are inherently similar, sharing V-matrices brings little  performance sacrifice.
> >
> > **(3)** Our U-matrix trimming serves the same **purpose of performance-efficiency tradeoff** as other expert pruning.  **Note that these trimmed U-matrices are derived from decomposable layer, which intrinsically contains more parameter redundancy and contributes slightly smaller to overall performance compared to the other layers**.
> >
> > **(4)** Our ablation in Table 4 shows that retaining the Top-2 U-matrix with the highest frequency and trimming the rest 6 U-matrices results in an ideal trade-off and greatly reduces the number of parameters.
> >
> > -----
> >
> > **Finally,** we hope these responses address the concerns, and we appreciate the constructive feedback. We are committed to improving our manuscript and believe the insights will significantly contribute to this goal. We are glad to discuss further comments and suggestions. If the reviewer finds our response adequate, we would really appreciate it if the reviewer considers raising the score sing the score.

---

### Author Response · Authors · 2024-11-22
**General Response**

**Dear Reviewers, Area Chairs, Senior Area Chairs and Program Chairs,**

We sincerely thank all reviewers for their positive feedback and constructive comments. Reviewers positively acknowledge **the novelty of the idea, the methodology employed, the extensive experiments conducted, the superior performance, and the good presentation of the paper**. More encouragingly, **Reviewer ghaz and eFn1** think our **novel framework tackles an important problem in MoE LLM efficiency** for the community:

**[Novelty]:**

- **Reviewer ghaz:** *"MoE-SVD introduces a unique method for MoE compression; the matrix-sharing techniques can be interesting."*
- **Reviewer eFn1:** *"Explores SVD-based methods for MoE LLMs and identifies key factors that degrade performance."*

**[Theoretical Soundness]:**

- **Reviewer eFn1:** *"Proposes a selective decomposition strategy along with low-rank matrix sharing and trimming, which are well-justified approaches."*
- **Reviewer nYc5:** *"The method seems technically sound and straightforward in principle."*

**[Comprehensive Evaluation]:**

- **Reviewer ghaz:** *"The paper thoroughly evaluates the proposed method across various MoE architectures, including Mixtral-8x7B, Phi-3.5-MoE, and DeepSeekMoE."*
- **Reviewer eFn1:** *"Additional experiments with LoRA fine-tuning and quantization validate the efficacy of MoE-SVD."*

**[Clear Presentation]:**

- **Reviewer eFn1:** *"Overall, the paper is clear and easy to follow."*
- **Reviewer nYc5:** *"Good written and easy to follow."*

**[Efficacy & Practicality]:**

- **Reviewer ghaz:** *"Additional experiments with LoRA fine-tuning and quantization validate the efficacy of MoE-SVD."*
- **Reviewer nYc5:** *"The method seems technically sound and straightforward in principle."*



In the past two weeks, we carefully improved the experiments (using all computational resources we have), the clarifications and the discussions of our work to address the concerns, the questions and the requests by all four reviewers. Summarily, we made the following improvements:

**(1)** To enhance the empirical validation of our MoE-SVD methodology, following Reviewers ghaz, eFn1, and nYc5, we have restructured our experimental results through six quantitative analyses in 6 tables: **(a)** Table 1 presents a rigorous representation of our decomposition strategy, replacing  the previous Figure 4; **(b)** Tables 11 and 12 demonstrate MoE-SVD's performance improvements (performance degradation of only 3% and 5%) on Mixtral-8×7B and Phi3.5 MoE under 20% compression after incorporating more calibration data, with statistical significance measurements.; **(c)** Tables 9 and 10 provide comparative analyses against established methods (Wanda, SparseGPT, LoSparse, MC-SMoE, and Unified-MoE) with key performance metrics (TFLOPs, runtime, model size) ; **(d)** Table 13 extends our evaluation to the Qwen 57B model; **(e)** A new analysis of Decomposed Layer/CKA distance justifies our matrix sharing strategy.

**(2)** To strengthen our methodological framework, responding to all reviewers, we have: **(a)** formalized our sensitivity score computation; **(b)** Clarify theoretical foundations for our V matrix sharing strategy through CKA distance analysis; **(c)** enhanced the methodology sections (lines 226-230, 243, 257) and appendices (B.2-B.4, C, D.1-D.2) with detailed explanations of sensitivity metrics, decomposition criteria, and implementation protocols.

**(3)** To elucidate the clarity of our approach, we have introduced three new algorithms tables that provide: **(a)** detailed implementations with PyTorch code for core components (sensitive scores and sampling frequency collection, experts decomposition, selection strategy); **(b)** refined pseudocode for the SVD-MOE framework and MakePositiveDefinite function (Algorithms 4-5).

**(4)** Following Reviewers nYc5 and eFn1, we have conducted runtime performance analyses on DeepSeekMoE and Qwen models in Table 10 and 13, validating our approach across diverse MoE architectures and shared expert configurations.





**Finally, we have carefully integrated all these improvements into a cohesive revision that we believe enhances the quality of our work.** We look forward to the reviewers' feedback on these comprehensive updates.





Best regards,

Paper 88 Authors

---

> ### Author Response · Authors · 2024-11-22
> **Key Addition Experiments Results**
>
> **Table A: Performance of MoE-SVD with 512 calibration samples under 20% compression ratios.**
>
> | Mixtral-8×7B        | WikiText-2↓     | PTB↓     | C4↓     | Openb.     | ARC_e     | WinoG.     | HellaS.     | ARC_c     | PIQA     | MathQA     | Average↑     |
> | ------------------- | --------------- | -------- | ------- | ---------- | --------- | ---------- | ----------- | --------- | -------- | ---------- | ------------ |
> | Original            | 3.98            | 12.99    | 6.78    | 0.36       | 0.84      | 0.76       | 0.65        | 0.57      | 0.82     | 0.43       | 0.63         |
> | MoE-SVD (256)       | 5.94            | 19.42    | 8.98    | 0.28       | 0.75      | 0.69       | 0.55        | 0.45      | 0.78     | 0.36       | 0.55         |
> | MoE-SVD (512)       | 4.44            | 15.21    | 8.32    | 0.32       | 0.78      | 0.73       | 0.57        | 0.48      | 0.79     | 0.37       | 0.58         |
> | MoE-SVD (512) +LoRA | 4.31            | 14.94    | 7.82    | 0.33       | 0.80      | 0.73       | 0.61        | 0.55      | 0.81     | 0.38       | 0.60         |
> | **Phi3.5 MoE**      | **WikiText-2↓** | **PTB↓** | **C4↓** | **Openb.** | **ARC_e** | **WinoG.** | **HellaS.** | **ARC_c** | **PIQA** | **MathQA** | **Average↑** |
> | Original            | 3.48            | 8.43     | 8.22    | 0.40       | 0.77      | 0.76       | 0.68        | 0.56      | 0.79     | 0.38       | 0.62         |
> | MoE-SVD (256)       | 4.77            | 12.12    | 9.56    | 0.39       | 0.77      | 0.69       | 0.59        | 0.53      | 0.74     | 0.35       | 0.58         |
> | MoE-SVD (512)       | 4.26            | 11.41    | 9.53    | 0.38       | 0.76      | 0.72       | 0.63        | 0.53      | 0.77     | 0.35       | 0.59         |
> | MoE-SVD (512) +LoRA | 4.29            | 10.99    | 8.81    | 0.39       | 0.81      | 0.74       | 0.65        | 0.54      | 0.79     | 0.36       | 0.61         |
>
>
>
> **Table B:  More comparisons with other structured compression methods under around 20% compression.**
>
> | Mixtral-8×7B (20%)   | Runtime Speed up | Runtime Token/Sec | WikiText-2↓ | PTB↓    | C4↓     | Openb. | ARC_e | WinoG. | HellaS. | ARC_c | PIQA | MathQA | Average↑ |
> | -------------------- | ---------------- | ----------------- | ----------- | ------- | ------- | ------ | ----- | ------ | ------- | ----- | ---- | ------ | -------- |
> | Wanda (2:4)          | 1.04x            | 91.1              | 4.72        | 18.8    | 8.43    | 0.32   | 0.76  | 0.72   | 0.55    | 0.47  | 0.79 | 0.36   | 0.57     |
> | SparseGPT (2:4)      | 1.06x            | 93.1              | 4.61        | 21.11   | 8.19    | 0.3    | 0.77  | 0.74   | 0.56    | 0.45  | 0.77 | 0.35   | 0.56     |
> | LoSparse             | 1.06x            | 92.9              | 953.51      | 805.16  | 1273.12 | 0.2    | 0.27  | 0.49   | 0.28    | 0.26  | 0.53 | 0.2    | 0.32     |
> | Merge, then Compress | 1.09x            | 95.3              | 1341.36     | 1316.52 | 1478.13 | 0.26   | 0.28  | 0.51   | 0.29    | 0.25  | 0.54 | 0.19   | 0.33     |
> | Unified-MoE-Compress | 1.13x            | 99.1              | 6.12        | 14.67   | 11.61   | 0.30   | 0.73  | 0.70   | 0.54    | 0.46  | 0.73 | 0.33   | 0.54     |
> | MoE-SVD         | 1.2x             | 104.7             | 4.44        | 15.21   | 8.32    | 0.32   | 0.78  | 0.73   | 0.57    | 0.48  | 0.79 | 0.37   | 0.58     |
>
>
>
> **Table C: The centered kernel alignment (CKA) similarity before/after decomposition for decomposed layer [3,5,6,7,9,12,23,24,25], in Mixtral-8×7B on WikiText-2.**
>
> | Decomposed Layer/CKA distance      | 3     | 5     | 6     | 7     | 9     | 12    | 23    | 24    | 25    |
> | ---------------------------------- | ----- | ----- | ----- | ----- | ----- | ----- | ----- | ----- | ----- |
> | Original Expert Matrix             | 0.279 | 0.334 | 0.335 | 0.341 | 0.338 | 0.329 | 0.273 | 0.269 | 0.21  |
> | Decomposed U-matrix of each Expert | 0.304 | 0.320 | 0.327 | 0.329 | 0.329 | 0.331 | 0.309 | 0.307 | 0.305 |
> | Decomposed V-matrix of each Expert | 0.998 | 0.984 | 1     | 1     | 1     | 0.989 | 0.978 | 0.999 | 0.983 |
>
>
>
> **Table D:  Performance of Qwen2-57B-A14B compressed by MoE-SVD under 20% compression ratios.**
>
> | Models                            | Runtime Token/Sec | WikiText-2↓ | PTB↓  | C4↓   | Openb. | ARC_e | WinoG. | HellaS. | ARC_c | PIQA | MathQA | Average↑ |
> | --------------------------------- | ----------------- | ----------- | ----- | ----- | ------ | ----- | ------ | ------- | ----- | ---- | ------ | -------- |
> | Qwen2-57B-A14B  Original          | 42.70             | 4.32        | 11.66 | 9.23  | 0.33   | 0.75  | 0.74   | 0.63    | 0.47  | 0.8  | 0.39   | 0.58     |
> | Qwen2-57B-A14B MoE-SVD (20%)      | 53.16(x1.25)      | 6.52        | 14.61 | 13.64 | 0.29   | 0.71  | 0.69   | 0.58    | 0.42  | 0.74 | 0.33   | 0.53     |
> | Qwen2-57B-A14B MoE-SVD (20%)+LoRA | 53.16(x1.25)      | 5.41        | 13.26 | 11.63 | 0.30   | 0.74  | 0.73   | 0.61    | 0.45  | 0.78 | 0.35   | 0.56     |

---

> ### Author Response · Authors · 2024-12-04
> **Addition Python pseudo-code for Understanding of Technical Details  (Part 1/2)**
>
> **Algorithm: Pyhon pseudo-code for sensitivity metric of MoE-SVD.**
>
> ```pseudocode
> def compute_layer_sensitivity(experts_weights, activations, gating_outputs, calibration_data, top_k=2, tau=2.0):
>     """
>     Compute layer-wise sensitivity metric S_L for MoE compression
>
>     Args:
>         experts_weights (list of torch.Tensor): Weight matrices for each expert
>         activations (list of torch.Tensor): Activation values for each expert
>         gating_outputs (torch.Tensor): Router outputs for calibration data
>         calibration_data (torch.Tensor): Calibration dataset
>         top_k (int): Number of experts to select per token
>         tau (float): Threshold for activation outliers
>
>     Returns:
>         float: Layer sensitivity score S_L
>     """
>     num_experts = len(experts_weights)
>     device = experts_weights[0].device
>
>     # Compute sampling frequency (f_i)
>     top_k_indices = torch.topk(gating_outputs, top_k, dim=-1).indices
>     expert_counts = torch.zeros(num_experts, device=device)
>     for indices in top_k_indices:
>         expert_counts[indices] += 1
>     f_i = expert_counts / len(calibration_data)
>
>     # Compute principal rank (r_i) using SVD
>     r_i = torch.zeros(num_experts, device=device)
>     for i, weight in enumerate(experts_weights):
>         U, S, V = torch.linalg.svd(weight)
>         # Count singular values above threshold
>         threshold = torch.max(S) * 1e-2  # Threshold
>         r_i[i] = torch.sum(S > threshold)
>
>     # Compute activation outliers (a_i)
>     a_i = torch.zeros(num_experts, device=device)
>     for i, activation in enumerate(activations):
>         mean_abs_act = torch.mean(torch.abs(activation))
>         outliers = torch.sum(torch.abs(activation) > tau * mean_abs_act)
>         a_i[i] = outliers / activation.numel()
>
>     # Compute final sensitivity metric S_L
>     S_L = torch.sum(f_i * r_i * a_i)
>
>     return S_L
>
> ```
>
> **Algorithm: Pyhon pseudo-code for SVD Expert Decomposition of MoE-SVD.**
>
> ```pseudocode
> class MoESVDCompression:
>     def __init__(self, truncate_k=None, top_k_experts=2):
>         """
>         Initialize Activation-Weighted SVD with Matrix Sharing and Trimming
>
>         Args:
>             truncate_k (int): Number of singular values to keep
>             top_k_experts (int): Number of top experts to select for U-matrix trimming
>         """
>         self.truncate_k = truncate_k
>         self.top_k_experts = top_k_experts
>
>     def compute_activation_weights(self, X):
>         """
>         Compute activation-weighted scaling matrix S using Cholesky decomposition
>
>         Args:
>             X (torch.Tensor): Activation matrix [batch_size,feature_dim]
>             torch.mm(X, X.t()) is Cumulative activation matrix, representing the sum of processed activation data.
>         """
>         # Compute Gram matrix
>         gram = torch.mm(X, X.t())
>
>         # Cholesky decomposition
>         S = torch.linalg.cholesky(gram)
>         return S
>
>     def decompose_expert(self, W_original, X):
>         """
>         Perform activation-weighted SVD on single expert
>
>         Args:
>             W_original (torch.Tensor): Original weight matrix
>             X (torch.Tensor): Activation matrix
>         """
>         # Compute activation-weighted matrix
>         S = self.compute_activation_weights(X)
>         W_aw = torch.mm(W_original, S)
>
>         # Perform SVD
>         U, sigma, V = torch.linalg.svd(W_aw, full_matrices=False)
>
>         # Truncate if specified
>         if self.truncate_k is not None:
>             U = U[:, :self.truncate_k]
>             sigma = sigma[:self.truncate_k]
>             V = V[:self.truncate_k, :]
>
>         return U, torch.diag(sigma), V, S
>
> ```

---

> ### Author Response · Authors · 2024-12-04
> **Addition Python pseudo-code for Understanding of Technical Details  (Part 2/2)**
>
> **Algorithm: Pyhon pseudo-code for Matrix Sharing \& Trimming of MoE-SVD.**
>
> ```pseudocode
> class MoESVDCompression:
>     def __init__(self, truncate_k=None, top_k_experts=2):
>         """
>         Initialize Activation-Weighted SVD with Matrix Sharing and Trimming
>
>         Args:
>             truncate_k (int): Number of singular values to keep
>             top_k_experts (int): Number of top experts to select for U-matrix trimming
>         """
>         self.truncate_k = truncate_k
>         self.top_k_experts = top_k_experts
>     def __init__(self, truncate_k=None, top_k_experts=2):
>         """
>         Initialize Activation-Weighted SVD with Matrix Sharing and Trimming
>
>         Args:
>             truncate_k (int): Number of singular values to keep
>             top_k_experts (int): Number of top experts to select for U-matrix trimming
>         """
>         self.truncate_k = truncate_k
>         self.top_k_experts = top_k_experts
>
>     def compress_moe(self, expert_weights, activations, routing_frequencies):
>         """
>         Compress MoE using V-matrix sharing and U-matrix trimming
>
>         Args:
>             expert_weights (list): List of expert weight matrices
>             activations (list): List of activation matrices for each expert
>             routing_frequencies (torch.Tensor): Expert selection frequencies
>         """
>         num_experts = len(expert_weights)
>         compressed_experts = []
>
>         # Decompose all experts
>         decomposed = []
>         for i in range(num_experts):
>             U, Sigma, V, S = self.decompose_expert(expert_weights[i], activations[i])
>             decomposed.append((U, Sigma, V, S))
>
>         # Select shared V-matrix based on highest routing frequency
>         max_freq_idx = torch.argmax(routing_frequencies)
>         V_shared = decomposed[max_freq_idx][2]
>
>         # Sort experts by routing frequency for U-matrix trimming
>         sorted_indices = torch.argsort(routing_frequencies, descending=True)
>
>         # Perform U-matrix trimming and construct compressed experts
>         for i in range(num_experts):
>             # Find top-k U-matrices from more frequently used experts
>             more_frequent = [j for j in sorted_indices if routing_frequencies[j] > routing_frequencies[i]]
>             top_k_indices = more_frequent[:self.top_k_experts]
>
>             if len(top_k_indices) < self.top_k_experts:
>                 # If not enough more frequent experts, use own U-matrix
>                 top_k_indices = top_k_indices + [i]
>
>             # Combine selected U-matrices and corresponding Sigma matrices
>             U_combined = torch.zeros_like(decomposed[i][0])
>             Sigma_combined = torch.zeros_like(decomposed[i][1])
>
>             for idx, expert_idx in enumerate(top_k_indices[:self.top_k_experts]):
>                 U_combined += decomposed[expert_idx][0]
>                 Sigma_combined += decomposed[expert_idx][1]
>
>             # Reconstruct compressed expert
>             W_compressed = torch.mm(torch.mm(U_combined, Sigma_combined),
>                                  torch.mm(V_shared, torch.inverse(decomposed[i][3])))
>             compressed_experts.append(W_compressed)
>
>         return compressed_experts
>
> ```

---

### Meta-Review · Area_Chair_QrpM · 2024-12-14

**Metareview:**

The submission proposes an SVD based method for mixture-of-experts compression.  As pointed out in the reviews, mixture-of-experts model compression has received a lot of attention with many competing methods.  The submission provides some improvements across the provided metrics and settings considered.  Two reviewers felt that the submission could be marginally acceptable, while one reviewer argued for rejection.  Due to some remaining concerns, a large number of competing methods, and weak support for acceptance, the submission falls below the threshold for acceptance to ICLR in its current form.

**Additional Comments On Reviewer Discussion:**

The main concerns of the reviewer recommending rejection were that the settings considered in the submission were not the most meaningful, and that the submission should compare to additional baselines.  The authors responded with some additional compression ratios, but argued that the baselines were "fundamentally different and not directly comparable." The revised submission contained additional compression ratios, but not these comparisons.  The reviewer was responsive to the author's concerns, but did not respond to their last messages, which prompted a strong negative response from the authors.  Nevertheless, the reviewer's concerns appear to be founded on legitimate concerns about evaluation settings, comparison to baselines, and coherent presentation.

---

### Decision · Program_Chairs · 2025-01-22

Reject